# Greenery-Covered Tall Buildings: A Review

**Kheir Al-Kodmany** 

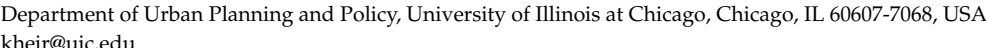

Department of Urban Planning and Policy, University of Illinois at Chicago, Chicago, IL 60607-7068, USA; kheir@uic.edu

**Abstract:** The greenery-covered tall building, an innovative building typology that substantially integrates vegetation into the design, promises to transform urban landscapes into more sustainable and livable spaces. This paper explores the concept of greenery-covered tall buildings. It achieves its objectives by offering an overall introduction to this building typology, mapping out novel projects to identify innovative ideas and design strategies, and reviewing the social, environmental, and economic benefits of integrating greenery into tall buildings. Examining prominent implementations distinguishes purposeful greenery integration from "afterthought" additions, providing insights for architects, developers, scholars, and the public. Additionally, the paper discusses the downsides and challenges of integrating trees and plants into tall buildings, including construction costs, maintenance considerations, and adherence to building and fire codes, and proposes remedies. This research fosters a deeper understanding of their transformative potential in creating greener, more resilient, and socially beneficial urban environments by contributing to the architectural discourse on this emerging building typology.

**Keywords:** high-rise buildings; vegetation; advantages; disadvantages; biophilic design; sustainability; greenwashing

## 1. Introduction

### 1.1. The Increasing Demand for Nature

People demand reconnecting with nature for multiple reasons, including massive urbanization, health problems, energy crises, artificial digital proliferation and increases in screen time, climate change and resilience, environmental and air quality degradation, the spread of the COVID-19 pandemic, and poor aesthetics and urban design. Densely populated cities are often characterized by high stress levels and noise pollution from traffic, construction, and other urban activities. Prolonged exposure to constant noise can lead to irritability and difficulty in relaxation and concentration. The constant presence of large crowds and congestion in urban areas can create a sense of claustrophobia and anxiety. Busy streets crowded public transportation, and packed public places can contribute to feeling overwhelmed [1–6].

Long periods of electronic device use have further increased the demand for reconnecting with nature. As modern life progressively revolves around electronic devices, such as smartphones, computers, and tablets, finding ways to disconnect from the digital world and reconnect with the natural world has become essential. Nature provides a respite from the constant stimuli of electronic devices and offers an opportunity to relax, unwind, and refresh mentally and physically. Outdoor activities like strolling through a park or relaxing in a green area can aid in lowering stress levels, elevate mood, and advance general well-being. Nature's calming effects can counteract the potential negative impacts of prolonged screen time and electronic device use [7,8].

Urban areas are often plagued by poor air quality, increasing the demand to integrate nature into urban living. The concentration of human activities, industries, transportation, and construction in urban settings leads to the emission of pollutants that can negatively impact air quality. Manufacturing plants and industrial facilities in urban areas release

various pollutants into the atmosphere, including sulfur dioxide ($SO_2$), carbon dioxide ($CO_2$), and other hazardous substances. Poor air quality can harm public health, leading to respiratory, cardiovascular, and other health complications. Buildings are significant sources of $CO_2$ emissions in urban environments (Figure 1). The energy used for lighting, cooling, heating, and operating appliances in residential, commercial, and institutional buildings often relies heavily on fossil fuels. The combustion of these fossil fuels, such as coal, natural gas, and oil, contributes to the greenhouse effect and climate change by releasing $CO_2$ into the atmosphere [9].

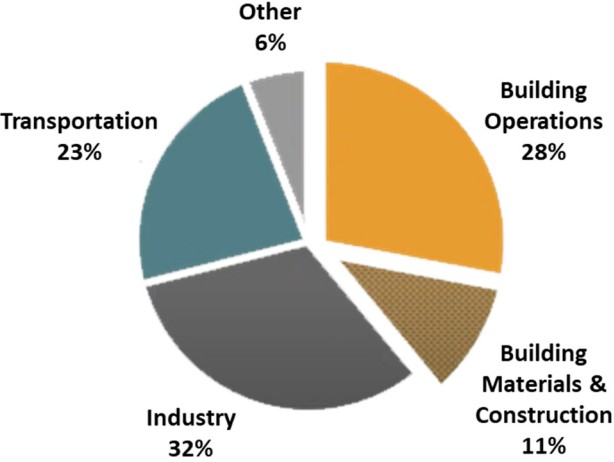

**Figure 1.** Global $CO_2$ Emissions by Sector. (Graph by author, adapted from [9]).

Recently, COVID-19 further emphasized the importance of cleaner air for public health and the need to reintegrate nature into cities. As a respiratory disease, COVID-19 has underscored the critical role of clean air in maintaining public health and well-being. Poor air quality, often prevalent in urban areas due to pollution, can exacerbate respiratory issues and weaken the immune system, making individuals more vulnerable to respiratory infections. This realization has prompted cities to reevaluate their urban planning and environmental policies, focusing on creating greener and more sustainable living spaces. The importance of reintegrating nature into cities has become evident during the pandemic [10,11].

Overall, the growing demand for reconnecting with nature stems from a recognition of the many benefits that nature offers in improving physical and mental health, promoting sustainability, and creating more pleasant and resilient urban environments. As cities face numerous challenges in the 21st century, integrating nature into the urban landscape has become a key priority for creating healthier, happier, and more sustainable places to live.

*1.2. Biophilic Design*

Biophilic design is a concept that recognizes the inherent human need to connect with nature and seeks to bridge the gap between the built environment and the natural world. By incorporating elements of nature into architectural and interior design, biophilic design aims to create spaces that evoke positive emotional responses and enhance the well-being of occupants. Integrating vegetation, plants, and greenery into buildings can profoundly impact the occupants' mental and physical health. Exposure to natural elements such as plants and natural light has been shown to reduce stress levels, improve mood, and increase productivity. Greenery and living plants in indoor spaces can also improve indoor air quality, as they absorb pollutants and release oxygen, creating a healthier and more pleasant indoor environment. Using natural sounds, such as running water or birdsong, and scents, such as the aroma of fresh flowers or wood, can also contribute to a more immersive and sensory experience, further strengthening the connection to nature.

Incorporating natural materials, such as wood and stone, can evoke a sense of grounding and connection to the natural world, promoting feelings of warmth and comfort.

Additionally, incorporating natural patterns and textures into the design can create a sense of harmony and balance, further enhancing the overall experience of the space. Biophilic design seeks to create spaces that provide functional utility and evoke a sense of well-being, tranquility, and connection to the natural world. By integrating natural elements and processes into the built environment, biophilic design offers an innovative and holistic approach to designing spaces that promote human health and resilience in the face of modern urban challenges [12–15].

### 1.3. Vertical Landscaping and Planting

Cities are getting denser, leaving little space for "horizontal" landscaping and planting. Consequently, architects have had to get creative in using the vertical plane for planting. They believe vertical green spaces can act as natural air purifiers, absorbing pollutants and improving air quality, making cities more resilient to public health crises. As such, they diligently explore vertical gardening, making lush vegetation and trees grow on the upper floors, terraces, balconies, walls, and roofs. They have been designing new projects that bring nature and gardens, usually found on the ground level, onto the high-rise building, allowing users to reconnect with nature and create natural environments in the sky [16,17].

Politicians may back up these projects for their merits. Increasingly, cities oppose all-glass skyscrapers because of their environmental harm. For example, Bill de Blasio, former New York City Mayor (from 2014 to 2021), has proposed a bill to ban all-glass skyscrapers to decrease NYC's greenhouse emissions by 30 percent [18]. According to de Blasio, all-glass towers are "incredibly inefficient" since so much energy escapes through the glass—they are the city's primary source of greenhouse gas emissions. Toronto, Canada, has been encouraging using timber framing—highly compressed wood, called cross-laminated timber wood, which is extremely strong—in constructing high-rise buildings [19,20]. Utrecht, the Netherlands, has required all buildings to have green or solar roofs [21]. In Singapore, the government supports structures that integrate greenery by covering up to half the cost. As a result, nearly all new buildings in Singapore are rich in vegetation. Many European cities are witnessing the proliferation of vertical greenery features such as living walls, vegetated terraces, and green roofs.

Italian architect Stefano Boeri invented the "vertical forest" concept, which involves constructing high-rise buildings extensively covered with trees and plants on various floors, creating a vertical ecosystem within urban environments. The core idea of the vertical forest is to bring nature back into urban centers by integrating lush greenery into tall buildings. Boeri demonstrated his concept in Milan, Italy, and "replicated" the model in different projects, including Trudo Vertical Forest in Eindhoven, the Netherlands; Easyhome Huanggang Vertical Forest City Complex in Huanggang, Hubei Province, China; and Nanjing Vertical Forest in Nanjing, Jiangsu Province, China. He has also proposed the same model for cities like Dubai [22].

Further, Boeri has proposed giant visions based on his original "vertical forest" model in Milan. He has suggested Liuzhou Forest metropolis as a model for a Chinese metropolis of around 1.5 million inhabitants in the hilly southern region of Guangxi, one of the most smog-affected urban areas in the world due to excessive industrialization and overpopulation. The concept depicts an urban composition along the Liujiang River that spans an area of 175 hectares and includes buildings such as offices, houses, hotels, hospitals, and schools that are almost totally contained by plants and trees of varying species and sizes. About 40,000 trees and one million plants representing over one hundred species will call the Liuzhou Forest City home. By integrating extensive trees and plants, Boeri's project aims to sequester significant quantities of carbon dioxide ($CO_2$) and microparticles while concurrently generating substantial amounts of oxygen to mitigate the detrimental effects of severe air pollution [23].

STH BNK by Beulah is a groundbreaking architectural project in Melbourne, Australia, that aims to become the world's first-ever supertall vertical garden. The development comprises two twisting towers, one rising to 366 m and the other to 288 m, connected by a

sky bridge. These towers will be situated above the Yarra River and will house a mix of residential, commercial, and retail spaces, a wellness hub, and a vertical school. With over five and a half kilometers of vertical gardens and sky parks extending as high as 365 m above street level, the project aspires to set a new standard for sustainable and green urban development. The towers' various levels will feature dramatic planting, with greenery adorning the building's façades, terraces, and sky parks. This extensive use of vegetation enhances the aesthetic appeal and contributes to environmental benefits like air purification and temperature regulation. The STH BNK project is designed by UN Studio and Cox Architecture, renowned architectural firms known for their innovative and sustainable design solutions. Construction has already commenced, and the project is expected to be completed by 2028 [24].

*1.4. Controversy*

As integrating greenery into tall buildings becomes a growing trend, growing controversy arises concerning this new building typology. The proponents of this architectural design approach claim the greenery-covered tower model offers multiple benefits, including improving the health of people and the environment and mitigating climate change challenges. Greenery-covered tall buildings have many benefits, such as purifying the air, reducing ambient temperature and noise, reducing stress, boosting productivity, and a longer residence time. Green coverings can significantly reduce other air pollutants, including soot and dust [25–27].

A growing concern is that this could be a new "greenwashing" propaganda. Architects are taking advantage of the positive public perception of plants and trees and their many health and environmental benefits by integrating trees and plants into a proposed building to "sell" their designs. However, adding plants and trees to tall buildings poses several challenges that the public (and even some professionals and scholars) may not be aware of [28]. Due to the additional structural requirements, irrigation systems, and specialized planting techniques required to support the weight of the vegetation, creating green spaces in tall buildings can be costly. Green areas in tall buildings require routine and specialized maintenance to ensure the health and vitality of the plants. This consists of pruning, watering, pest control, and monitoring for prospective problems. Incorporating vegetation into structures necessitates compliance with fire and building codes, which may have specific requirements for fire resistance, egress routes, and safety measures. Green spaces within towering buildings may affect the space's functionality. For instance, windows and views may be obstructed, and interior floor plans may need to be modified to accommodate the natural elements. Maintaining green spaces in lofty buildings frequently necessitates centralized management and control to ensure uniform care and a consistent appearance. This can inhibit the organic and spontaneous growth typically observed in natural environments [29,30].

## 2. Objectives

This paper aims to explore and delve into the unique building typology of greenery-covered tall buildings, which involves integrating vegetation into the design of tall structures. At its core, the paper stimulates a qualitative architectural discourse, encouraging readers to engage in thoughtful discussions and reflections on this innovative concept. By highlighting critical issues surrounding greenery-covered tall buildings, the paper fosters a deeper understanding and appreciation of the potential and challenges associated with this emerging architectural model. Exploring greenery-covered tall buildings opens up a world of possibilities for architects, developers, scholars, and urban planners. By investigating the integration of vegetation into vertical structures, the paper inspires creative thinking and sparks curiosity about the sustainable, aesthetic, and societal implications of such projects.

At the idea level, the paper invites readers to contemplate the harmony between nature and urban environments. It prompts discussions on the positive impact of greenery on the well-being of inhabitants, the potential contributions to ecological sustainability, and the

enhancement of urban aesthetics. It also encourages critical examination of the challenges that must be addressed, such as water management, maintenance, and compatibility with existing urban infrastructure. Through this qualitative architectural discourse, the paper encourages stakeholders to envision the future of urban landscapes, where greenery-covered tall buildings play a pivotal role in shaping environmentally responsible and socially conscious cities. By fostering meaningful conversations around this novel typology, the paper catalyzes further research, exploration, and development in sustainable architecture and urban planning.

The key objectives are as follows:

1. Raise Awareness and Introduction: By offering an overall introduction to greenery-covered tall buildings, the paper aims to raise awareness among architects, developers, scholars, and the general public about this novel building typology. This introduction is a starting point for understanding the principles, benefits, and potential of integrating greenery into tall buildings.

2. Mapping Innovative Projects: The paper explores and highlights innovative ideas and design concepts by mapping novel projects that integrate greenery. The paper showcases successful implementations that can serve as transformative architectural solutions. By reviewing these projects, the readers can differentiate them from "afterthought" greenery additions.

3. Review Social, Environmental, and Economic Benefits: The paper aims to comprehensively review the many benefits of integrating greenery into tall buildings. This examination delves into how such projects positively impact the well-being of urban residents, contribute to sustainability, and enhance the urban environment.

4. Addressing Challenges: The paper intends to discuss the challenges of integrating trees and plants into tall buildings. These challenges include construction costs, maintenance considerations, and compliance with building and fire codes. Understanding these obstacles is essential for developing successful and sustainable greenery-covered tall buildings.

The paper seeks to contribute valuable knowledge to the architectural discourse by accomplishing these objectives, inspiring further research, innovation, and responsible urban development. It encourages stakeholders to consider the potential of greenery-covered tall buildings in shaping more sustainable, vibrant, and nature-oriented urban environments. Moreover, the paper offers practical insights into overcoming challenges, promoting well-informed decision making, and fostering a deeper understanding of this emerging architectural model.

## 3. Methods

Therefore, this study aims to evoke a qualitative discussion at the idea level by conducting a literature review to examine the innovative concepts applied in greenery-covered tall buildings. Literature sources are increasingly abundant as the online environment is flourishing. To meet the study's goals, this paper reviews vast sources of information about the topic, including academic literature, architectural magazines, websites, blogs, documentaries, and videos. It also uses the Council on Tall Buildings and Urban Habitat (CTBUH) Skyscraper Center's database, one of the most comprehensive tall buildings databases.

In conducting the review, there must be a process and criteria for selecting projects for examination. Here are the primary criteria:

1. Extensive Greenery Coverage: The building should showcase a significant amount of greenery covering a substantial portion of its envelope, such as vegetated balconies, terraces, or vertical gardens. In the case of tall buildings, the roof area is proportionately much smaller than the entire building's envelope. As such, if a tall building has implemented only a green roof, it does not fit the "greenery-covered tall building" typology and is not selected for examination.

2. Integration of Greenery in Design Scheme: The greenery should be an integral part of the building's design scheme, reflecting a deliberate effort towards sustainability rather than a superficial addition after the building's design was finalized. The greeneries should not just sprinkle a few trees and shrubs on the building or an afterthought scheme. It is essential to go beyond superficial additions and ensure that greenery is thoughtfully incorporated from the early stages of the design process. This deliberate effort towards sustainability ensures that the green elements are aesthetically pleasing but also functional and beneficial.

3. Completed Construction and Inhabitation: The project must be completed, construction finished, and the building should be inhabited. Numerous proposed and visionary projects show massive integration of greeneries into buildings. However, these projects stay on the drawing board. Completed projects offer more practical insight than visionary ones.

4. Minimum Height Requirement: The building should be 10+ stories tall, ensuring that the study focuses on tall buildings embodying the challenges and benefits of greenery in high-rise structures. Please see Appendix A to discern the rationale.

After examining the architecture literature while applying the above criteria, the study identified thirty-one projects. Table 1 lists these projects chronologically to trace the development and evolution of this building typology across the globe. It summarizes each project concisely, indicating the name, location, architect, number of floors, function, year of completion, distinctive vegetative features, and thumbnail image.

**Table 1.** Major Greenery-Covered Tall Buildings. (Compiled by author).

| # | Building Name | Location | Architect | Num. of Floors | Function | Year of Compl. | Distinctive Vegetative Features | Thumbnail Image |
|---|---|---|---|---|---|---|---|---|
| 1 | Consorcio Building | Santiago, Chile | Enrique Browne and Borja Huidobro | 17 | Office | 1993 | -Vegetative skin on the outer side of the western façade, articulated in three recessed vegetative bands of 4 floor-heights for each [31] |  |
| 2 | ACROS Fukuoka Prefectural International Hall | Fukuoka, Japan | Emilio Ambasz | 17 | Civic Center | 1995 | -Fifteen stepped, vegetated terraces with mature trees connected via stairs and spraying jets of water<br>-Green roof [32] |  |
| 3 | The Met | Bangkok, Thailand | WOHA | 36 | Residential and Hotel | 2005 | -Vegetated balconies<br>-Sky gardens<br>-Sky terraces [33] |  |

**Table 1.** *Cont.*

| # | Building Name | Location | Architect | Num. of Floors | Function | Year of Compl. | Distinctive Vegetative Features | Thumbnail Image |
|---|---|---|---|---|---|---|---|---|
| 4 | Council House 2 (CH2) | Melbourne, Australia | City of Melbourne | 10 | Office | 2006 | -Vegetated façades articulated via metal meshes that hold plants and connect balconies vertically [34] |  |
| 5 | Newton Suites | Singapore | WOHA | 36 | Residential | 2007 | -Rooftop planting -Vegetated balconies -Green walls [35] |  |
| 6 | School of the Arts Singapore (SOTA) | Singapore | WOHA | 10 | Educational | 2009 | -Green walls -Vertical strips of plants along the building's façades -Green roofs [36] |  |
| 7 | Khoo Teck Puat Hospital | Singapore | CPG Consultants in collaboration with RMJM | 10 | Hospital | 2010 | -Green terraces -Green roofs [37] |  |
| 8 | One Central Park | Sydney, Australia | Jean Nouvel | 34, 14 | Residential | 2013 | -Façade-supported green walls [38] |  |
| 9 | CDL's Tree House | Singapore | CDL | 24 | Residential | 2013 | Green walls cover more than 2300 square meters of the building's façades [39] |  |

**Table 1.** *Cont.*

| # | Building Name | Location | Architect | Num. of Floors | Function | Year of Compl. | Distinctive Vegetative Features | Thumbnail Image |
|---|---|---|---|---|---|---|---|---|
| 10 | Bosco Verticale, "Vertical Forest" | Milan, Italy | Stefano Boeri Architetti | 27, 19 | Residential | 2014 | -Balconies are placed in a staggered pattern, allowing trees to grow up multiple floors [40] |  |
| 11 | CapitaGreen | Singapore | WOHA | 16 | Office | 2014 | -Vegetation wraps around the building's perimeter -Considerable indoor plants [41] |  |
| 12 | Santalaia | Bogotá, Columbia | Exacta Proyecto Total | 11 | Residential | 2015 | -Vertical garden of more than 115,000 plants of 10 different species, covering an area of 3117 square meters [42] |  |
| 13 | East Village | Beirut, Lebanon | Jean Marc Bonfils and Associates | 12 | Residential | 2015 | -Vertical garden that covers the façade of one of the blocks [43] |  |
| 14 | M6B2 Tower of Biodiversity | Paris, France | Maison Edouard François | 18 | Residential | 2016 | -The tower's titanium cladding creates moiré patterns that give it a subtle and fluctuating appearance [44] |  |

**Table 1.** *Cont.*

| # | Building Name | Location | Architect | Num. of Floors | Function | Year of Compl. | Distinctive Vegetative Features | Thumbnail Image |
|---|---|---|---|---|---|---|---|---|
| 15 | Oasia Downtown | Singapore | WOHA | 27 | Office | 2016 | -Steel exoskeleton that holds plants along the entire height of the building<br>-Vegetated sky courts [45] |  |
| 16 | ParkRoyal on Pickering | Singapore | WOHA | 16 | Hotel and Office | 2016 | -Large terraces that contain extensive plants, tall trees, and water features<br>-Green walls<br>-Plants along the building's perimeter [46] |  |
| 17 | Le Nouvel KLCC | Kuala Lumpur, Malaysia | Ateliers Jean Nouvel | 49, 43 | Residential | 2016 | -Vegetated façade<br>-Living walls<br>-Green roof [47] |  |
| 18 | Clearpoint Residencies | Colombo, Sri Lanka | Arosha Perera | 47 | Residential | 2017 | -Garden terraces for each apartment along the entire height of the tower [48] |  |
| 19 | Huaku Sky Garden | Taipei, Taiwan | WOHA | 38 | Residential | 2017 | -Vegetative balconies<br>-Plants that grow along vertical concrete screens [49] |  |
| 20 | Kampung Admiralty | Singapore | WOHA | 11 | Residential/ Mixed Use | 2017 | -Sky parks<br>-Community plazas [50] |  |

**Table 1.** *Cont.*

| # | Building Name | Location | Architect | Num. of Floors | Function | Year of Compl. | Distinctive Vegetative Features | Thumbnail Image |
|---|---|---|---|---|---|---|---|---|
| 21 | The Tao Zhu Yin Yuan | Taipei, Taiwan | Vincent Callebaut | 21 | Residential | 2017 | -Vegetated balconies and terraces (trees, shrubs, and plants) are incorporated on each floor, wrapping the entire tower [51] |  |
| 22 | Torre Rosewood | São Paulo, Brazil | Jean Nouvel | 22 | Hotel | 2018 | -Vertical garden -Terraces and rooftops -Flowers, plants, and trees [52] |  |
| 23 | Check Point | Tel Aviv, Israel | Itamar Lensky and Noa Zuckerman | 12 | Office | 2019 | -Living wall that covers more than 80% of its exterior surface -Green roof that collects rainwater [53] |  |
| 24 | Qiyi City Forest Garden Tower 4 | Chengdu, China | Chengdu Qiyi Real Estate Co., Ltd. | 30 | Residential | 2019 | -Balcony in every unit containing lush plants -The vegetation scheme consists of 20 types of plants that create a vertical forest effect [54] |  |
| 25 | "1000 Trees" | Shanghai, China | Heatherwick Studio | 10 | Mixed Use | 2019 | -Vertical planters that contain trees and a mixture of plants also have the structural function of holding the building together [55] |  |
| 26 | Sky Green Residential & Retail Tower | Taichung City, Taiwan | WOHA | 26 | Mixed Use | 2019 | -Protruding balconies with plants and trees -Sky gardens -Trellises for green creeper plants [56] |  |

| # | Building Name | Location | Architect | Num. of Floors | Function | Year of Compl. | Distinctive Vegetative Features | Thumbnail Image |
|---|---|---|---|---|---|---|---|---|
| 27 | Eden | Singapore | Heatherwick Studio | 26 | Residential | 2020 | -Curved balconies overflow with trees, shrubs, and plants along the entire height of the tower [57] |  |
| 28 | Trudo Vertical Forest | Eindhoven, Netherlands | Stefano Boeri Architetti | 19 | Residential | 2021 | -Protruding balconies that incorporate extensive plants, trees, and shrubs. They are placed in a staggered pattern, allowing tall trees to grow up multiple floors [58] |  |
| 29 | Easyhome Huanggang Vertical Forest City Complex | Huanggang, Hubei Province, China | Stefano Boeri Architetti | 28 | Mixed Use | 2022 | -Protruding balconies that incorporate extensive plants, trees, and shrubs. They are placed in a staggered pattern, allowing tall trees to grow up multiple floors [59] |  |
| 30 | Ravel Plaza | Amsterdam, Netherlands | MVRDV | 29, 23, 19 | Residential | 2020 | -Vegetation covers the residential units' balconies, terraces, roofs, and façades [60] |  |
| 31 | Nanjing Vertical Forest | Nanjing, Jiangsu Province, China | Stefano Boeri Architetti | 35, 18 | Residential | 2023 | -Two towers that host a variety of plants on their balconies, creating a vertical forest that covers an area of 4500 square meters [61] |  |

Ideally, it would be valuable to examine all the listed buildings. However, due to the space limit in a single article, the author had to select a few projects for examination. The study aims to investigate tall buildings that incorporate distinctive or "innovative" vegetative concepts. It looks at various elements related to greenery integration in these buildings, listed in Table 1. These elements include:

1.  Vegetated Balconies: Tall buildings with vegetative balconies feature greenery or plantings on their outdoor balconies, providing residents with a connection to nature and enhancing the building's aesthetics.
2.  Terraces and Rooftop Gardens: Buildings with vegetated terraces incorporate green spaces or gardens on elevated platforms, creating outdoor spaces for relaxation and recreation while promoting biodiversity.
3.  Façade-Supported Green Walls: They use elements (e.g., wires, cables, netting, lattice, or mesh) to support climbing plants, allowing them to spread and grow along the building's façade.
4.  Vegetated Exoskeleton: The exoskeleton is an external support structure that may also be designed to accommodate greenery and vegetation, contributing to the building's sustainability and aesthetics.
5.  Façade-Integrated Green Walls: Façade-integrated green walls are a type of living wall system where the plants, substrate, and structural support are directly attached to the building wall. They may involve modular panels or containers pre-planted with various plant species and mounted on the wall.
6.  Vertical Planters: Buildings with vertical planters incorporate specially designed containers or structures to house plants and vegetation, often installed on the building's exterior.

By examining these elements in tall buildings, the study seeks to understand how innovative vegetative concepts are integrated into urban architecture and how they contribute to sustainability, aesthetics, and the overall urban environment. The research may shed light on best practices and inspire future architectural designs that embrace greenery and enhance the quality of life in urban settings. Table 2 reorganizes the listed buildings in Table 1 based on their prime vegetative elements.

**Table 2.** Greenery-covered high-rises based on the dominant vegetative feature. (Compiled by author).

| | Dominant Distinctive, Innovative Vegetative Features | Buildings |
|---|---|---|
| 1 | Vegetated balconies | Bosco Verticale<br>The Met<br>Newton Suites<br>CapitaGreen<br>Clearpoint Residencies<br>Qiyi City Forest Garden Tower 4<br>Huaku Sky Garden<br>Torre Rosewood<br>The Tao Zhu Yin Yuan<br>Eden<br>Trudo Vertical Forest<br>Sky Green Residential & Retail Tower<br>Easyhome Huanggang Vertical Forest City Complex<br>Ravel Plaza<br>Nanjing Vertical Forest |
| 2 | Terraces and rooftop gardens | ACROS Fukuoka Prefectural International Hall<br>Khoo Teck Puat Hospital<br>ParkRoyal on Pickering<br>Kampung Admiralty |

**Table 2.** *Cont.*

| | Dominant Distinctive, Innovative Vegetative Features | Buildings |
|---|---|---|
| 3 | Façade-supported green walls (wires, cables, netting, or lattice/mesh) | One Central Park<br>Consorcio Building<br>Council House 2 (CH2)<br>M6B2 Tower of Biodiversity<br>Le Nouvel KLCC |
| 4 | Exoskeleton | Oasia Downtown |
| 5 | Façade-integrated green walls | CDL's Tree House<br>Santalaia<br>East Village<br>Check Point<br>School of the Arts Singapore (SOTA) |
| 6 | Vertical planters | "1000 Trees" |

The next step involved focusing on one representative building from each of the six categories, a pragmatic approach that responds to the limit of what a single article can cover while still providing valuable insights into the subject matter. By selecting one representative building from each category, the research can showcase a diverse range of greenery-covered tall buildings and explore their unique characteristics, design concepts, challenges, and benefits. Each selected case study can act as an exemplar, representing its respective typology and providing valuable information for the overall research.

To help in choosing just one representative for each category, an additional criterion concerning receiving awards and recognitions was applied. Accolades indicate that the projects have been acknowledged and appreciated by experts and professionals in the field, further validating their innovative design and sustainable features. By focusing on projects with accolades and certifications, the research can highlight exemplars of greenery-covered tall buildings that have excelled in sustainability and have been celebrated for their positive contributions to the urban landscape. These projects can serve as role models and inspire other architects and developers to incorporate greenery into their designs thoughtfully and effectively.

The following explains the selection process for buildings representing each category.

1. Vegetated Balconies. Vegetated balconies represent one of the most popular concepts in greenery-covered tall buildings. They offer a practical and accessible way to integrate greenery into the building's design while providing numerous benefits for occupants and the environment. Vegetated balconies offer a direct connection to nature for building occupants. Residents can have their private green spaces, allowing them to enjoy the benefits of nature, such as improved well-being and stress reduction, without having to leave their homes. Table 2 illustrates that almost half of the 31 listed projects use vegetative balconies. Among these projects, Bosco Verticale in Milan, designed by Stefano Boeri, stands out as a pioneering example that has garnered worldwide interest and acclaim. The innovative use of vegetative balconies in Bosco Verticale has sparked a new wave of interest in integrating greenery into building typologies, making it a crucial case study for this research. The project has received worldwide recognition and is considered an exemplary greenery-covered tall building. It is a must-study case.

2. Terraces and Rooftop Gardens. Spacious vegetative terraces and rooftop gardens are distinctive features of greenery-covered tall buildings. The ACROS Fukuoka Prefectural International Hall, completed in 1995 in Fukuoka, Japan, stands as one of the earliest and most remarkable examples of greenery-covered tall buildings. The building's unique design incorporates a stepped garden façade, creating a series of terraces that rise from ground level to the rooftop. These terraces are filled with lush

vegetation, creating a visually stunning and environmentally friendly addition to the urban landscape. This building's innovative design has garnered appreciation and admiration over the years, proving its longevity and relevance in sustainable architecture. The building's design is a testament to the seamless integration of architecture and nature, dedicating significant areas to accommodate substantial greenery. Therefore, it is selected as a representative case study.

3. Façade-Supported Green Walls. One Central Park is a remarkable example of a greenery-covered tall building, exemplifying the successful integration of vegetation and innovative design. The project has received considerable praise for its innovative approach to urban greenery and sustainability, mainly due to the design team of famous architect Jean Nouvel and landscape architect Patrick Blanc. The green walls of One Central Park feature a diverse selection of plant species that add color and texture to the urban landscape. These vertical gardens enhance the building's aesthetic appeal and contribute to several sustainable features.

Jean Nouvel and Patrick Blanc have extended their innovative vegetative façade system to a larger scale with the Le Nouvel KLCC project in Kuala Lumpur, Malaysia. Le Nouvel KLCC comprises twin towers, with one tower containing 49 floors and the other 43 floors. The project represents another impressive example of greenery-covered tall buildings, showcasing how the integration of vegetation can be successfully applied to urban architecture on a grand scale. Like One Central Park, the vegetative façade system in Le Nouvel KLCC involves lush vertical gardens that adorn the building's exterior. These green walls add a distinctive and aesthetically pleasing element to the towers and contribute to various sustainable features. The above two buildings could be ranked as "tied", and both will be examined.

4. Vegetated Exoskeleton. The concept of a "vegetative exoskeleton" is indeed an innovative and original design approach, expanding on the idea of lattice structures by enveloping the entire building with greenery, as in the case of Oasia Downtown. The building's distinctive design and sustainable and biophilic features have earned it architectural recognition and acclaim. Oasia Downtown, designed by WOHA Architects, exemplifies sustainable architecture that seamlessly integrates nature into its urban setting. The vegetative exoskeleton enhances the building's visual appeal and is an effective environmental solution. The green façade acts as a natural sunshade, providing passive cooling for the building's interior and reducing energy consumption. The building's numerous accolades reflect its success as a model for sustainable and biophilic architecture.

5. Façade-Integrated Green Walls. The concept of façade-integrated green walls is another innovative and distinctive feature of greenery-covered high-rises, showcasing the seamless integration of vegetation into the building's walls. Several notable buildings, such as the School of the Arts Singapore (SOTA) in Singapore, Santalaia in Bogotá, Colombia, and Check Point in Tel Aviv, Israel, have embraced this design approach, creating visually striking and environmentally friendly structures.

CDL's Tree House in Singapore is a remarkable example of applying the concept of façade-integrated green walls with greater height, showcasing how greenery can wrap around the entire building surface. This design enhances the building's aesthetic appeal and contributes to its environmental sustainability and biophilic qualities. CDL's Tree House's recognition through various awards, including the BCA Green Mark Platinum Award, FIABCI Singapore Property Awards, Skyrise Greenery Award, and Southeast Asia Property Awards, underscores its success as a sustainable and nature-oriented architecture model.

6. Vertical Planters. Vertical planters are an innovative concept in greenery-covered tall buildings, and "1000 Trees" in Shanghai, China, stands as a unique example of their application. The building's design revolves around the idea of incorporating vertical planters, showcasing how vegetation can be creatively integrated into the

building's façade. The vertical planters act as a prominent visual element, giving the building its distinctive and environmentally friendly appearance. Overall, "1000 Trees" in Shanghai is a compelling example of vertical planters in greenery-covered tall buildings, and exploring its design can provide valuable knowledge for furthering the understanding of this innovative building typology. Table 3 lists the seven selected case studies with their associated distinctive, innovative vegetative features.

**Table 3.** The seven examined case studies, their geographic locations, years of completion, and distinctive vegetative features.

| | Buildings | Location | Year of Completion | Dominant Distinctive/Innovative Vegetative Features |
|---|---|---|---|---|
| 1 | Bosco Verticale | Milan, Italy | 2014 | Vegetated balconies |
| 2 | ACROS Fukuoka Prefectural International Hall | Fukuoka, Japan | 1995 | Terraces and rooftop gardens |
| 3 | One Central Park | Sydney, Australia | 2013 | Façade-supported green walls (wires, cables, netting, or lattice/mesh) |
| 4 | Le Nouvel KLCC | Kuala Lumpur, Malaysia | 2016 | |
| 5 | Oasia Downtown | Singapore | 2016 | Exoskeleton |
| 6 | CDL's Tree House | Singapore | 2013 | Façade-integrated green walls |
| 7 | "1000 Trees" | Shanghai, China | 2019 | Vertical planters |

## 4. Case Studies

### 4.1. Bosco Verticale, Milan, Italy

Bosco Verticale (Vertical Forest) is pioneering and the most representative of the greenery-covered high-rise building model. Its boldness, significant height, and extensive greeneries have promoted this project among the most remarkable. Bosco Verticale was Stefano Boeri's invention. It is considered a revolutionary project—a model for a sustainable residential building—that sparked and inspired other projects worldwide. Located in Milan, Italy, Bosco Verticale comprises two residential towers (27 and 19 stories, respectively 112 and 80 m high).

Completed in 2014, the main feature of the towers is their vast, staggered, overhanging balconies, each around three meters long. These spacious balconies fit massive outdoor tubs for vegetation and permit the growth of larger trees—giving the towers a forest-like appearance (Figure 2). As such, the buildings integrate 800 trees, each standing 3, 6, or 9 m (10, 20, or 30 feet) tall, and a diverse assortment of shrubs and flowers. Plants total about 15,000 perennials and ground-covering plants and 5000 bushes. The project offers 30,000 m² of woodland on a 3000 m² footprint [23,40,62–65].

Years after its completion, Bosco Verticale established a habitat inhabited by various animal species, including more than 1600 bird and butterfly specimens. This established an outpost for the metropolitan area's natural recolonization of flora and fauna. The plant-based façades filter the Sun's rays, creating a comfortable indoor microclimate. Greeneries generate oxygen, absorb $CO_2$ and microparticles, and "regulate" humidity. In addition, the project uses an energy system that produces electricity by employing photovoltaic panels. The demand for irrigation is also centralized; a "smart" system tracks the needs of the plants [66].

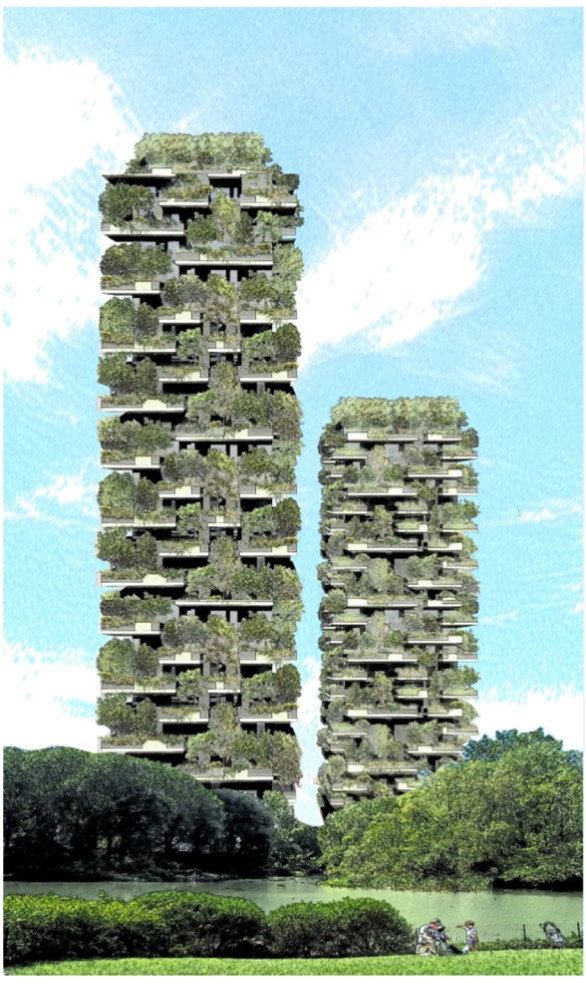

**Figure 2.** Bosco Verticale (Vertical Forest) in Milan, Italy. The towers' façades feature extensive vegetation. (Sketch by author).

The plants' colors and shapes create an iridescent landmark visible from afar in every season. This trait has made Milan's Vertical Forest a new symbol in just a few years. The choice of plants and trees on the towers' sides and floors reflects aesthetic and practical requirements to adapt to the façades' orientations and heights. The botanical part resulted from three years of research with botanists and ethnologists. It started in the summer of 2010 when the plants that would go on the towers were grown in a special botanical "nursery" near Peverelli Nursery and garden center to get them used to living in conditions like those in their natural habitat [67].

Bosco Verticale, an architectural marvel in the heart of Milan, Italy, redefines the urban landscape by introducing an exquisite fusion of nature and modernity. Translating to "Vertical Forest", this iconic building stands as a living testament to sustainable design, innovation, and the profound connection between humans and the natural world. Bosco Verticale is more than a building; it is a sanctuary that beckons us to reimagine our relationship with the cities we inhabit. It ignites a conversation about the transformative power of blending modernity with nature, revitalizing our urban centers while championing sustainable coexistence. This towering forest stands tall as an emblem of hope and a reminder that architecture can catalyze change, where innovation, beauty, and ecological mindfulness come together harmoniously.

Since its completion, Bosco Verticale has received several recognitions, certifications, and awards for its innovative and sustainable design. Some of the most notable ones are:

- The 2014 International Highrise Award;
- The 2015 Best Tall Building Worldwide by the Council on Tall Buildings and Urban Habitat (CTBUH);
- The 2015 CTBUH Urban Habitat Award (in addition to winning the Best Tall Building Worldwide Award, Bosco Verticale also received the CTBUH Urban Habitat Award in 2015);
- The 2018 Emporis Skyscraper Award.

*4.2. ACROS Fukuoka Prefectural International Hall, Fukuoka, Japan*

Located in the heart of the city of Fukuoka, Japan, the ACROS Fukuoka Prefectural International Hall is a hub for international, cultural, and informational interaction. It aimed to provide an innovative solution to a prevalent urban issue: combining a developer's desire for lucrative site use with the public's need for open green space. By developing an innovative agro-urban model, Fukuoka's strategy satisfies both objectives.

Designed by pioneering green architect Emilio Ambasz, the Asian Cross Roads Over the Sea (ACROS) is a 17-story civic center building that was completed in 1995. The building integrates nearly 100,000 square meters of park space onto fifteen stepped, vegetated terraces with staircase-shaped rooftops that ascend the entire structure's height [32,68,69]. Each terrace floor features a variety of gardens for contemplation, relaxation, and retreat from the city's congestion (Figure 3).

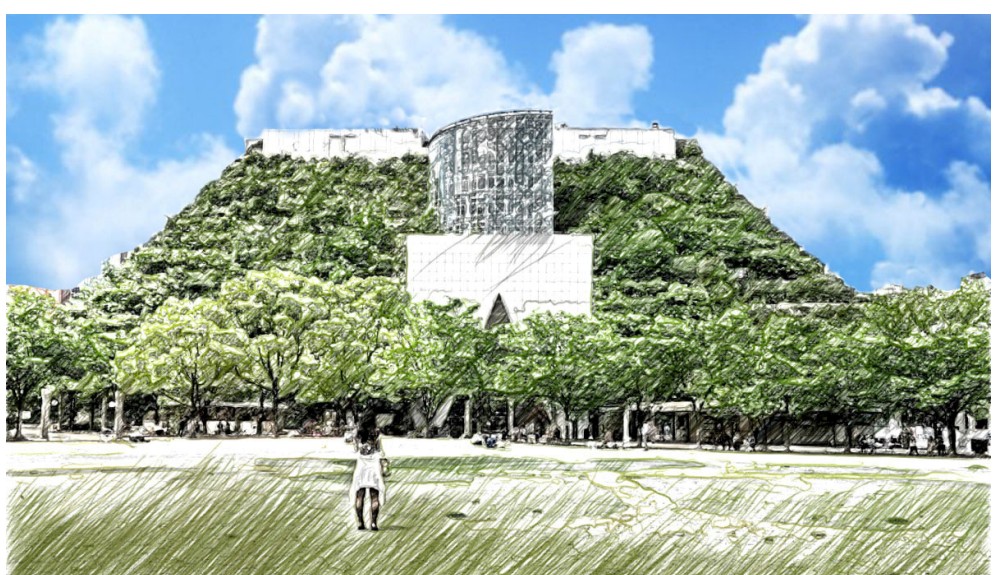

**Figure 3.** ACROS Fukuoka Prefectural International Hall in Fukuoka, Japan. (Sketch by author).

The terraces are connected by upward-spraying water jets to create a climbing waterfall resembling a ladder to conceal the ambient city noise. Open to the public, the building culminates with a magnificent rooftop observation deck, providing a breathtaking view of the bay of Fukuoka, the mountains, and the surrounding natural landscape. With spacious green roof terraces, the building provokes the image of a lush green mountain that extends into the adjacent park—making the park and the building inseparable. Since the project's construction, it has become a new landmark for the city [70].

Growing medium depth ranges from 12 to 24 inches. Initially, there were 76 plant varieties comprising 37,000 plants. After 25 years, birds carried in seeds and increased the step garden's plant life to 120 species and 50,000 plants [71]. On the garden's top, tenth, sixth, and fifth levels, longwave and shortwave radiation meters, ultrasonic three-dimensional wind speed and temperature meters, and scintillometers were installed to capture data on the thermal environment. The study found a 15 °C difference between the surface

temperatures of exposed surfaces and plant-covered areas, concluding that vegetation inhibits an increase in ambient air temperature [72].

The ACROS Fukuoka Prefectural International Hall in Fukuoka, Japan, presents an awe-inspiring embodiment of architectural innovation seamlessly intertwined with natural elements. This iconic structure is characterized by its striking terraced façade, resembling a lush hillside rather than a conventional building. The terraces cascade downward, creating an intricate blend of greenery and urbanity. Each terrace hosts a meticulously curated garden, blending native vegetation with urban landscapes, and together they form a captivating "step garden" effect that integrates the natural with the built environment. Its living façade not only captivates the eyes but also redefines the relationship between urban development and the natural world. ACROS serves as an inspiration, reminding us that harmonizing built spaces with nature can elevate the quality of our surroundings and contribute to a more sustainable and harmonious future.

ACROS Fukuoka Prefectural International Hall has received several recognitions and awards for its innovative and sustainable architecture. Some of them are:

- The 1995 Nikkei New Office Building Award;
- The 1996 Architectural Institute of Japan (AIJ) Annual Architectural Design Commendation;
- The 1997 Fukuoka City Architecture Award;
- The 2012 Asian Townscape Award.

*4.3. One Central Park, Sydney, Australia*

Located in Sydney, Australia, and designed by Jean Nouvel, One Central Park (OCP) (or Block 2) comprises two residential towers (34 and 14 stories) that were completed in 2013. A network of cascading planted terraces connects the tower to the nearby park. The building's landscape vertically extends the planted area of the nearby urban park, providing occupants with extraordinary living space and representing a potent green symbol on Sydney's skyline. About 50% of the building's façade is covered with a vertical landscape created in partnership with French botanist and artist Patrick Blanc [38].

OCP's façades are vertical gardens with over 200 native Australian plant species that grow from planters on every floor (Figure 4). Thirty-eight thousand exotic and native plants are packed within hydroponically nourished felt-faced panels. These plants were chosen for their climatic tolerance, durability, and beauty. Plants create a "musical composition" consisting of buds, vegetation and vine blooms, and leafy foliage that climbs on walls and metal strings. In addition to outstanding aesthetics, the "green veil" offers essential functions, including shade and purifying the air—plants sequester carbon dioxide, emit oxygen, and absorb heat. The plants serve as a natural solar control system that adapts to the seasons, protecting the apartments from direct sunlight in the summer and letting in the most sunshine possible in the winter [73,74].

Noticeably, plants are irrigated via recycled water supplied by the on-site stormwater collection tanks and the central blackwater treatment plant. A computerized irrigation system adds minerals and fertilizers to water based on data from sensors and weather stations—informing about plants' needs and microclimatic conditions (humidity, wind, temperature, and sunlight).

OCP's summit is crowned with a hovering cantilever full of mirrors. In conjunction with heliostat motorized mirrors (which track the Sun's movement) placed on the roof of the shorter building, the system captures sunlight and directs rays downward to dark spaces between the two towers.

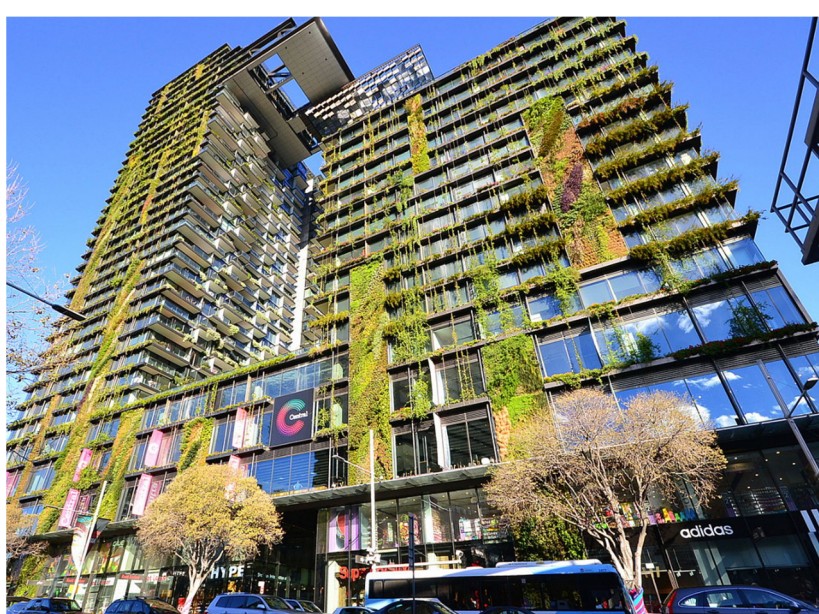

**Figure 4.** One Central Park, Sydney, Australia. (Credit: By Sardaka (talk) 08:28, 8 July 2014 (UTC)—Own work, CC BY 3.0, https://commons.wikimedia.org/w/index.php?curid=33832728 (accessed on 18 August 2023)).

One Central Park is a testament to the transformative power of weaving nature into urban architecture. Its vertical gardens, sky-high oasis, and lush podium demonstrate how sustainable design can elevate the quality of urban life, foster environmental stewardship, and inspire a new paradigm of urban living. This iconic high-rise is a beacon, reminding us that integrating green elements into urban spaces is vital to creating vibrant, resilient, and harmonious future cities. Adorning the façades of the towers are intricate vertical gardens, living tapestries that cascade down the buildings in a breathtaking display. These botanical masterpieces serve as stunning visual focal points and carry profound environmental significance. They act as natural air purifiers, absorbing pollutants and emitting oxygen, improving air quality in the bustling urban environment.

Due to a combination of sustainable design features, Block 2 is the first residential tower in Sydney to earn a 6 Green Star rating [73]. Other recognitions the building received include:

-   2014 Council on Tall Buildings and Urban Habitat (CTBUH) Best Tall Building Award—Asia & Australasia;
-   2014 International Highrise Award;
-   2014 Australian Institute of Architects (AIA) National Architecture Awards—Jørn Utzon Award for International Architecture;
-   2014 Asia Pacific Property Awards—Best Residential High-Rise Development;
-   2014 World Architecture Festival (WAF)—World Building of the Year.

### 4.4. Le Nouvel KLCC, Kuala Lumpur, Malaysia

Le Nouvel KLCC is a luxury residential development located in the heart of Kuala Lumpur, Malaysia. The project comprises two towers of 49 and 43 stories, totaling 195 units. Jean Nouvel, a renowned French architect known for his innovative and distinctive style, designed it. The towers offer a range of amenities and facilities for its residents, such as a sky lounge, a fitness center, a swimming pool, a jacuzzi, a sauna, a steam room, and a landscaped garden. The development is situated in the heart of Kuala Lumpur's city center, near the iconic Petronas Twin Towers and KLCC Park [47,74].

The architectural design of the towers is distinguished by their unique and innovative green façades, featuring climbing plants, intricate geometric patterns, and dynamic lines

(Figure 5). The façades add aesthetic appeal and serve practical functions such as sunshading and optimizing natural light. Incorporating greenery contributes to the well-being of the occupants, fostering a connection with nature while enjoying the convenience of urban living. The towers boast lush vertical gardens and a few terraces, creating a refreshing and serene environment within the bustling urban setting. The use of eco-friendly materials, energy-efficient technologies, and green building practices align with contemporary standards for sustainability and contribute to reducing the project's ecological footprint [75].

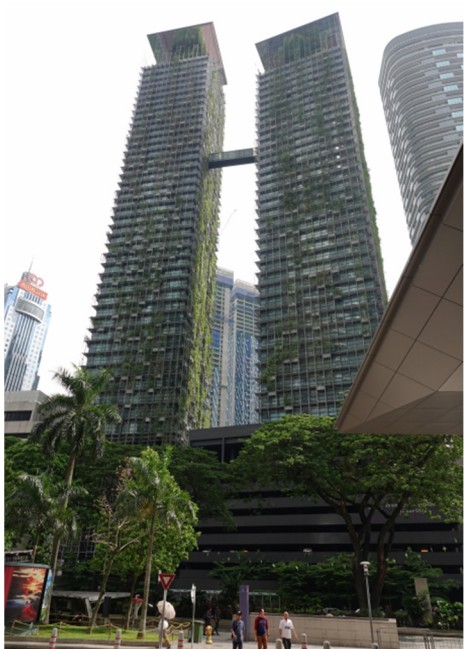

**Figure 5.** Le Nouvel KLCC, Kuala Lumpur, Malaysia. (Photo by author).

Le Nouvel KLCC stands as a symbol of architectural excellence and a testament to the harmonious integration of modernity and nature. With its innovative design, extensive greenery, and strategic location, the development exemplifies a new paradigm of urban living, offering a balance of luxury, sustainability, and convenience. The building's exterior is a symphony of vertical gardens, terraces, and balconies adorned with various plant life. This innovative use of green spaces contributes to the aesthetic appeal and has profound environmental implications. The living façade acts as a natural air filter, absorbing pollutants and emitting fresh oxygen, contributing to improved air quality in the urban environment.

The project has received significant awards, including:

- The 2018 International Highrise Award;
- The 2018 FIABCI Malaysia Property Award.

*4.5. Oasia Downtown, Singapore*

The Oasia Hotel Downtown is a lush tower of green in the middle of Singapore's congested Central Business District (CBD). It serves as a model for land use intensification in tropical cities. This "living tower" presents an alternative to the sleek and sealed skyscrapers that sprang out of the West. Oasia Downtown is a 27-story (190 m) office building that was completed in 2016.

As an integral aspect of the development's internal and external material palette, landscaping is used extensively, creating a haven for avian and mammalian wildlife and supporting biodiversity in the urban environment with a green plot ratio of 1100% [45]. The tower's red aluminum mesh covering is meant to serve as a backdrop, emerging from behind the 21 different species of creepers that cover it. These creepers provide nectar and pollen for the birds and insects that live in the area [76,77]. The building's peak is not a flat

surface but a tropical bower full of flowers and other soft and vibrant plants. Vegetation provides shade, which helps absorb heat, maintain a comfortable temperature, and clean the air (Figure 6).

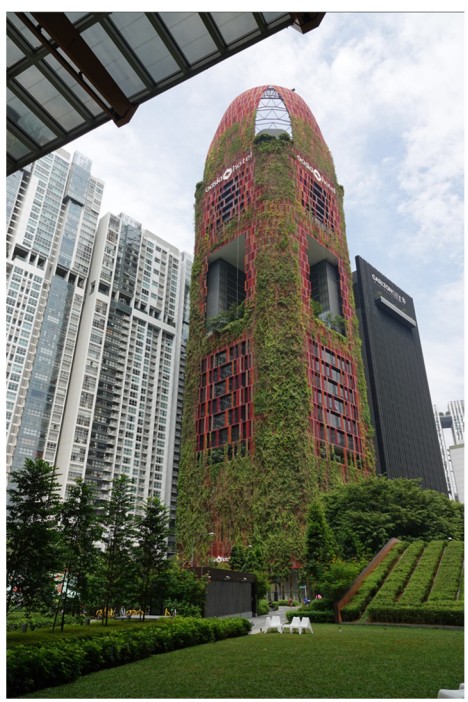

**Figure 6.** Oasia Downtown in Singapore. (Photo by author).

Each sky garden is designed as a verandah on an urban scale, shielded at a high level by the sky garden that precedes it and with open sides for visual transparency. To keep the tower cool, the architects interspersed open-air sky gardens with the vegetal façade at 30 m (98 ft) intervals and incorporated several energy-efficient fans. As a result, instead of being contained in internalized air-conditioned spaces, the public areas become practical, comfortable, tropical environments with plants, natural light, and fresh air [77].

In summary, Oasia Downtown in Singapore is a remarkable testament to innovative architectural design harmoniously blended with nature. The building's distinctive greenery-clad exterior exudes visual allure and serves as a functional ecosystem, contributing to the urban environment's health and vitality. The incorporation of extensive greenery, from ground-level landscaping to towering vertical gardens, showcases the building's commitment to sustainability and biophilic design principles. This living exterior acts as a natural air purifier, absorbing pollutants and carbon dioxide while releasing oxygen into the atmosphere. This enhances the air quality and cultivates a serene and refreshing ambiance within the bustling cityscape. As a vertical oasis in the city's heart, Oasia Downtown transcends traditional architectural norms, breathing life into the urban landscape while embodying a harmonious coexistence between nature and human innovation.

The Oasia Downtown building has received outstanding awards for its innovative and green design. Some of them are [51]:

- The 2016 Skyrise Greenery Awards;
- The 2016 CTBUH Urban Habitat Award;
- The 2018 FIABCI Singapore Property Award.

*4.6. CDL's Tree House, Singapore*

The Tree House, designed by CDL, is a residential development in Singapore that showcases a variety of innovative green features and technologies. One of the most striking aspects of the project is the plants and vegetation scheme, which covers more than

2300 square meters of the building's façade. The plants and vegetation scheme consists of several layers and types of plants, carefully selected and arranged to create a harmonious and sustainable living environment. The main components of the scheme are:

- The green wall is the scheme's most significant and visible part, covering the four sides of the 24-story tower (Figure 7). The green wall comprises modular panels containing various plants, such as ferns, orchids, bromeliads, and vines. The panels are irrigated by a rainwater-harvesting system and monitored by sensors to ensure optimal growth conditions. The green wall reduces the surface temperature of the building by up to 3 °C and lowers the energy consumption for cooling by up to 15% [39].
- The sky gardens are landscaped terraces on every four tower floors, providing residents access to green spaces and views. The sky gardens feature different themes and plant species, such as tropical, edible, medicinal, and aromatic gardens. The sky gardens also serve as communal areas for social interaction and recreation.
- The roof garden is the highest level of the scheme, covering the entire roof area of the tower. The roof garden consists of a lawn, a pavilion, a playground, and a jogging track. The roof garden offers panoramic views of the surrounding landscape, cityscape, outdoor activities, and relaxation opportunities. The roof garden also reduces the heat island effect and stormwater runoff [39].

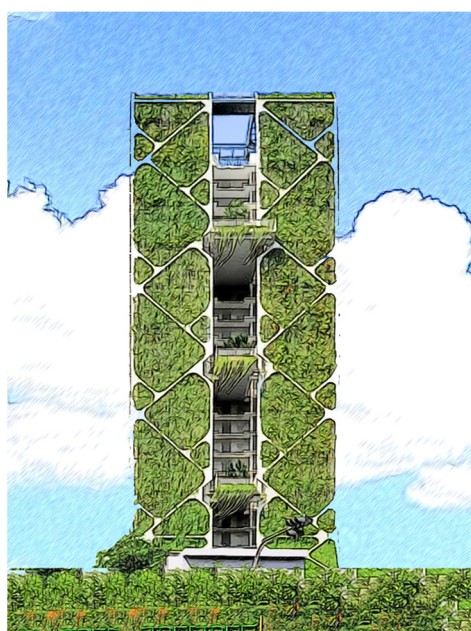

**Figure 7.** CDL's Tree House, Singapore. (Sketch by author).

From a comprehensive perspective, integrating vertical vegetation into the architectural design enhances its visual appeal and provides numerous environmental benefits. These factors include the reduction of heat accumulation, the improvement of air quality, and the promotion of biodiversity. The integration of nature and architecture in this context provides aesthetic satisfaction and serves as an active agent in promoting the welfare of the environment and its occupants.

The project has received several awards, including:

- The 2014 Building and Construction Authority (BCA) Green Mark Platinum Award;
- The 2014 Singapore Institute of Architects (SIA) Architectural Design Awards—Honorable Mention;
- The 2015 BCA Green Mark Platinum Award;
- The 2015 FIABCI Prix d'Excellence Awards—Silver Winner.

### 4.7. "1000 Trees", Shanghai, China

"1000 Trees", also known as "1000 Trees Plaza", is a mixed-use development in Pudong District, Shanghai, China. The complex integrates retail spaces, office areas, cultural venues, and public areas to create a vibrant, dynamic environment that encourages interaction and engagement. Designed by Heatherwick Studio, the project was completed in 2019.

The building represents a unique and innovative "green" architectural concept. The most distinctive feature of "1000 Trees" is its undulating exterior façade, which appears like a mountain range covered in vegetation (Figure 8). The façade is adorned with over 1000 large planters, each accommodating a tree or various types of plants. In addition to the tree-filled planters, the building incorporates extensive vertical gardens that cascade down the façades, further enhancing the greenery and contributing to the project's sustainability [55].

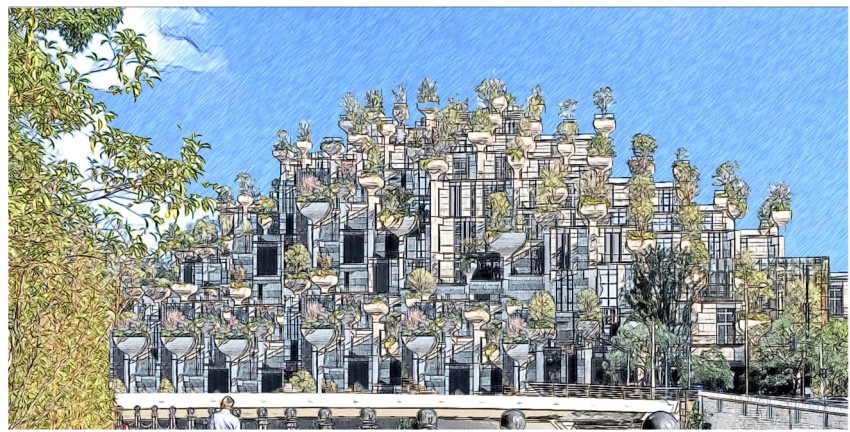

**Figure 8.** "1000 trees", Shanghai, China. (Sketch by author).

The abundant greenery provides a visually striking appearance and helps improve air quality, regulate indoor temperatures, and mitigate the urban heat island effect. The landscape design of the complex is carefully curated to complement the architecture and provide an oasis-like environment amidst the bustling urban setting of Shanghai.

The "1000 Trees" building has become a prominent emblem of excellence in Shanghai's urban landscape, representing the city's steadfast commitment to sustainable urban development and innovative architectural design. This statement shows the limitless potential for integrating green sanctuaries into heavily crowded urban areas, promoting a harmonious coexistence between nature and modernization. This development is a notable illustration of how inventive design may effectively integrate with environmental awareness, leading to a balanced and rejuvenating environment that enhances residents' well-being and the metropolitan scenery [55].

The project has received several awards, including:

- The 2021 CTBUH Award of Excellence for Best Tall Building 100–199 m;
- The 2020 Blueprint Award for Best Non-Public Project: Commercial;
- The 2019 MIPIM Asia Award for Best Futura Mega Project;
- The 2018 LEAF Award for Best Future Building Under Construction.

## 5. Discussion

Greenery-covered tall buildings offer benefits and face challenges. The following discussion dwells on ten benefits and ten challenges.

### 5.1. Benefits

The discourses around the reviewed projects, building designers, and developers emphasize the benefits of incorporating green elements into high-rise structures, creating a positive perception. The advantages of greenery-covered tall buildings span environ-

mental, social, and economic dimensions, rendering them an appealing and sustainable architectural paradigm for urban development [78–80]. The subsequent subsections strive to provide a concise overview of the asserted advantages, presented in a list of ten points. Compiling a succinct enumeration of the purported advantages associated with incorporating vegetation within high-rise structures might yield a simple and comprehensive outline of the benefits.

The following advantages highlight the diverse benefits of integrating vegetation into high-rise structures. The points and subpoints exhibit interrelations, interconnectedness, and overlapping. While each individual point has merits, the interconnected nature of these benefits contributes to a holistic and comprehensive impact on urban development and the well-being of humans and the environment. In other words, the advantages encompass a broad spectrum of benefits, showcasing the intricate web of connections between them. While each point holds its distinct value, it is essential to recognize that these benefits are interwoven, forming a cohesive and synergistic impact on urban development and the overall well-being of humans and the environment. By considering these advantages collectively, we gain a comprehensive understanding of the transformative potential that greenery-covered tall buildings hold in reshaping our cities for a more sustainable and harmonious future.

### 5.1.1. Improved Air Quality

Greenery-covered tall buildings can improve air quality through natural filtration and pollutant absorption mechanisms [81,82]. The breakdown below elucidates this process:

- Natural Filtration: Vegetation on building roofs, façades, and within urban spaces acts as a natural filter for pollutants present in the air. Various airborne particles get trapped on the surfaces as air passes through the leaves and vegetation [1,2].
- Carbon Dioxide ($CO_2$) Reduction: Greenery-covered tall buildings play a vital role in absorbing carbon dioxide, a significant greenhouse gas responsible for global warming and climate change. Through photosynthesis, plants absorb $CO_2$ and convert it into oxygen and glucose. This process helps reduce $CO_2$ levels in the atmosphere and releases oxygen into the air, enhancing air quality [1,2].
- Fine Particulate Matter (PM2.5) Capture: Fine particulate matter, often called PM2.5, consists of tiny particles suspended in the air. When inhaled, these particles can adversely affect human health, causing respiratory problems and other health issues. The leaves and surfaces of greenery-covered buildings act as surfaces where PM2.5 particles can settle, effectively reducing their concentration in the air [6,9].
- Healthier Urban Environments: By capturing and sequestering $CO_2$ and PM2.5, green façades contribute to cleaner and healthier air in the surrounding urban environment. This is particularly important in densely populated cities where air pollution levels can become hazardous due to increased human activities and pollution sources [17].
- Indoor Air Quality Improvement: Greenery-covered buildings also positively impact indoor air quality. The natural filtration process extends indoors, helping to reduce indoor air pollutants and creating healthier living and working environments for building occupants [17].
- Reduced Energy Consumption (see Section 5.1.5): Additionally, green façades provide thermal insulation, reducing the need for excessive air conditioning and heating. This leads to lower energy consumption, reducing emissions from energy production, and improving air quality [16,35].

Overall, greenery-covered tall buildings offer a multifaceted approach to improving air quality. They act as natural filters, absorbing $CO_2$ and capturing pollutants like PM2.5, resulting in cleaner and healthier air for building occupants and the public. This integration of nature into urban spaces showcases the potential of sustainable architecture in enhancing air quality and overall well-being.

### 5.1.2. Urban Heat Island Mitigation

Urban heat island mitigation is another potential advantage of tall vegetation-covered structures. The urban heat island effect occurs when cities experience higher temperatures than rural areas, predominantly due to human activities, dense infrastructure, and heat-absorbing materials such as concrete and asphalt [25,26,83]. The following breakdown provides a clear explanation of this process:

- Natural Shade and Cooling: Green façades, rooftops, and terraces of tall buildings provide natural shade, preventing direct sunlight from heating up building surfaces. The vegetation is a protective layer that blocks significant solar radiation from reaching the building's exterior. This natural shading reduces the heat the building absorbs, lowering its surface temperature [1,2,26].
- Evapotranspiration: Plants undergo transpiration, releasing water vapor into the air. This process helps cool the surroundings as water evaporates heat from the environment. As a result, the temperature in and around greenery-covered tall buildings can be notably lower than in non-vegetated areas [25].
- Evaporative Cooling: The combination of transpiration and evaporation leads to a cooling effect known as evaporative cooling. The moisture released by plants absorbs heat from the environment, effectively reducing ambient temperatures. This cooling effect is especially beneficial during hot and sunny days when urban heat is more intense [27].
- Microclimate Creation: As a result, incorporating greenery into tall buildings allows architects and urban planners to create microclimates within urban spaces. These microclimates offer more excellent and more comfortable conditions for occupants and pedestrians, contrasting with the otherwise elevated temperatures in urban areas [35].

Greenery-covered tall buildings can mitigate the urban heat island effect by providing natural shade, facilitating evaporative cooling, and creating cooler microclimates. Reducing surface temperatures and energy demand for cooling contribute to a more sustainable and comfortable urban environment for building occupants and the public.

### 5.1.3. Biodiversity Support

Greenery-covered tall buildings contribute significantly to supporting biodiversity in urban environments [12,13,84,85]. The explanation of this process is provided below:

- Habitat Creation: Including vegetation on building façades, rooftops, and terraces creates new habitats that can host a variety of plant and animal species [85].
- Insect-Friendly Environment: Greenery-covered tall buildings attract insects like bees, butterflies, and other pollinators. These insects play a crucial role in pollinating plants, contributing to the reproduction of various plant species and the overall health of ecosystems [85].
- Food Source: Vegetation provides a potential food source for urban wildlife, including birds and small mammals. Fruits, seeds, and insects attracted to the greenery offer nourishment for diverse species [12].
- Native Plant Species: Native plant species for green façades and rooftops support local ecosystems by providing food and shelter for native insects, birds, and pollinators. Native plants are well adapted to the local climate and often require less maintenance [84].
- Corridor for Movement: Greenery-covered tall buildings can serve as stepping stones or corridors for wildlife movement in urban areas. These corridors help connect fragmented habitats, allowing animals to move more freely and access essential resources [85].
- Enhanced Resilience: Diverse plant species in greenery-covered buildings can contribute to ecosystem resilience. In the face of climate change and environmental challenges, biodiversity-rich areas are better equipped to adapt to and withstand stressors [3,15].

- Educational Opportunities: Greenery-covered tall buildings provide educational opportunities for residents and visitors to learn about local plant and animal species. This increased awareness fosters a sense of responsibility for urban biodiversity conservation [48].

In general, greenery-covered tall buildings support biodiversity by creating new habitats, providing food sources, and attracting various plant and animal species. These buildings promote urban biodiversity, contributing to healthier ecosystems and more sustainable urban environments.

### 5.1.4. Enhanced Well-Being

Greenery-covered tall buildings can improve human well-being by providing physical, psychological, and social benefits [10,11,25,86,87]. The following outlines some of these benefits:

- Improved Air Quality (see Section 5.1.1): As discussed earlier, the vegetation on greenery-covered tall buildings acts as a natural filter, absorbing pollutants such as carbon dioxide ($CO_2$) and fine particulate matter (PM2.5). Cleaner air leads to better respiratory health and well-being for building occupants and the surrounding community [1,2,86].
- Stress Reduction: Greenery has been shown to reduce stress and anxiety. Exposure to natural elements, even in an urban environment, can have a calming effect, promoting mental well-being. Views of green spaces have been linked to increased feelings of happiness and reduced mental fatigue [19,25].
- Physical Activity: Accessible green spaces encourage physical activity, such as walking or relaxation. Rooftop gardens and green terraces provide opportunities for exercise and relaxation, promoting a healthier lifestyle [5].
- Temperature Regulation (see Section 5.1.6: Greenery-covered tall buildings contribute to cooler microclimates within urban areas. The shading effect of vegetation helps regulate temperatures, providing comfort for occupants and reducing heat-related stress [70,75].
- Noise Reduction (see Section 5.1.7): Vegetation can act as a buffer against noise pollution, creating a quieter environment for building occupants. Reduced noise levels contribute to better concentration and mental well-being [87].
- Social Interaction: Green spaces within tall buildings provide areas for social interaction and community engagement. Shared rooftop gardens or terraces create opportunities for residents to connect, fostering a sense of belonging and community [10,11].
- Engaging Sensory Experiences: Residents and visitors can engage with the greenery through sensory experiences such as smelling the fragrance of flowers, feeling the texture of leaves, and observing wildlife. These experiences add depth and engagement to the urban setting [88].
- Aesthetic Value (Section 5.1.9): The visual beauty of greenery-covered tall buildings enhances the overall aesthetics of the urban landscape. Beautiful and well-designed green spaces can evoke positive emotions and improve the overall ambiance of the area [40].

Overall, greenery-covered tall buildings can contribute to human well-being by improving air quality, reducing stress, fostering a connection to nature, promoting physical activity, regulating temperatures, reducing noise, facilitating social interactions, and offering aesthetic and educational benefits. These buildings could create healthier and more livable urban environments for residents and visitors alike.

### 5.1.5. Energy Efficiency

Greenery-covered tall buildings may offer advantages when it comes to boosting energy efficiency. These innovative designs incorporate vegetation and sustainable features that reduce energy consumption and promote a more environmentally friendly built environment [21,30,75,87]. The following details this notion:

- Natural Insulation: The vegetation on building façades, rooftops, and terraces acts as a natural insulator, providing an additional layer of thermal protection. This insulation helps regulate indoor temperatures, reducing the need for excessive heating during colder months and cooling during hotter periods [21].
- Reduced Heat Gain: Greenery-covered surfaces absorb solar radiation, reducing the heat entering the building. This mitigation of heat gain lessens the need for air conditioning and artificial cooling systems, leading to energy savings [30].
- Shading Effect: The strategic placement of greenery provides shading to building surfaces, including windows and walls. This shade prevents direct sunlight from reaching indoor spaces, minimizing overheating and the demand for cooling systems [86].
- Evaporative Cooling: Plants release water vapor into the air through transpiration, creating a cooling effect. This natural evaporative cooling can help reduce indoor temperatures and reliance on energy-intensive air conditioning systems [75].
- Improved HVAC Efficiency: The moderating effect of greenery on temperature and humidity levels can lead to more efficient operation of heating, ventilation, and air conditioning (HVAC) systems. These systems do not have to work as hard to maintain desired indoor conditions [30].
- Reduced Carbon Emissions: As energy consumption decreases due to energy-efficient design features, the associated carbon emissions from energy generation also decrease. This contributes to overall sustainability and a smaller carbon footprint [21].
- Renewable Energy Integration: In line with green design, greenery-covered tall buildings can integrate solar panels, wind turbines, or other technologies. These systems harness natural resources to generate power, reducing reliance on conventional energy sources [75].

In general, the incorporation of greenery on tall buildings has the potential to improve energy efficiency through various means. These include the provision of natural insulation, which helps to reduce heat gain, as well as the provision of shading and evaporative cooling. Additionally, integrating greenery on tall buildings can promote efficiency in heating, ventilation, and air conditioning (HVAC) systems and create a healthier urban environment. These structures exemplify a comprehensive approach to sustainable architecture that aligns with contemporary environmental considerations and objectives for energy conservation.

### 5.1.6. Protection of Building Structure

Greenery-covered tall buildings can feature valuable protection to the building structure by integrating vegetation and green features. This innovative approach not only enhances the aesthetic appeal of the building but also contributes to its long-term durability and resilience [25–27]. The explanation of this process is provided below:

- Protection from UV Radiation: The leaves and branches of plants provide a natural shield against the Sun's harsh ultraviolet (UV) radiation. UV rays can cause material degradation, surface fading, and building material deterioration. The vegetation absorbs and filters UV radiation, extending the lifespan of building elements [25].
- Regulating Temperature. Greenery-covered façades and roofs can help regulate temperature fluctuations on the exterior surfaces of a building. This can reduce tension and strain on the building envelope, thereby preventing potential problems caused by thermal expansion and contraction [27].
- Prevention of Weathering: Green façades, particularly those with climbing plants, shield the building façade from direct exposure to weather elements such as heavy rain, wind, and sunlight. This protection prevents premature weathering and deterioration of building surfaces [26].
- Reduced Wind Impact: Tall buildings are often exposed to strong winds that exert considerable force on their façades and structures. The presence of vegetation can act as a buffer, absorbing wind energy and reducing the impact on residents sitting on balconies and terraces, for example [87].

- Enhanced Resilience: The combination of natural elements and built structures creates a more resilient building that can withstand various environmental challenges, such as extreme weather events and climate change impacts [15].

In a broad sense, vegetation on tall buildings can safeguard the structural integrity of the buildings through various mechanisms. These include temperature regulation, waterproofing capabilities, mitigation of UV radiation, attenuation of wind forces, erosion control, noise reduction, enhancement of air quality, prevention of weathering, improvement of aesthetic appeal, and overall enhancement of resilience. The aforementioned advantages illustrate how incorporating vegetation on tall structures promotes the long-term durability of the structure.

### 5.1.7. Noise Reduction

Greenery-covered tall buildings can reduce noise pollution by acting as natural sound barriers and absorbing sound waves [25,26,87,88]. In densely populated urban areas, noise pollution originates from vehicular traffic, construction, and other human activities. Greenery-covered tall buildings situated strategically along busy streets or in noisy neighborhoods can intercept and absorb some of these noises, reducing their impact on residents. The following details this premise:

- Sound Absorption: Structures of plants, especially leaves, stems, and branches, have inherent sound-absorbing qualities. Sound waves are partially absorbed rather than reflected when they hit these surfaces. This absorption process helps decrease noise intensity, making the environment quieter [87].
- Diffusion: Vegetation on greenery-covered tall buildings can help scatter and disperse sound waves in multiple directions. This diffusion process prevents sound waves from concentrating in one direction and reduces the direct impact of noise on building occupants [25].
- Background Noise: Greenery introduces a layer of background noise, which can mask or cover up unwanted noises from the surroundings. The rustling of leaves, the chirping of birds, and the gentle flow of water in features like fountains create a more serene acoustic environment [27].
- Frequency Attenuation: Different vegetation types affect different sound frequencies. Dense vegetation with a mix of plant types can attenuate a wide range of frequencies, helping to reduce noise pollution from various sources [87].
- Strategic Design: Proper urban planning and landscape design can optimize the placement of greenery-covered tall buildings to strategically shield noise-sensitive areas from noise sources, such as busy roads or industrial zones [88].

Overall, greenery-covered tall buildings can reduce noise by absorbing, diffusing, and attenuating sound waves, creating a buffer between indoor spaces and noisy surroundings and providing a more serene acoustic environment. This noise reduction enhances the quality of life for building occupants and contributes to the overall well-being and livability of the urban environment.

### 5.1.8. Rainwater Management

Greenery-covered tall buildings may offer practical solutions for managing rainwater through a process known as "green infrastructure" [87,89,90]. Green infrastructure acts as a "sponge", absorbing rainwater and gradually releasing it into the atmosphere. This approach mimics natural water cycles and helps balance water resources and the environment. The explanation of this process is provided below:

- Stormwater Retention: The vegetation on greenery-covered tall buildings captures and retains rainwater on various surfaces, such as leaves, stems, and soil. This retention helps reduce the immediate volume of stormwater runoff that would otherwise flow into storm drains and potentially cause flooding [91].

- Rainwater Absorption: Plants can absorb rainwater through their roots and release it into the atmosphere through transpiration. This natural cycle helps regulate the amount of rainwater that enters the drainage system and decreases the strain on overburdened sewer systems [87].
- Evapotranspiration: Transpiration and evaporation from the soil and plant surfaces contribute to evapotranspiration. This process effectively reduces the overall amount of rainwater runoff by returning water to the atmosphere instead of overwhelming drainage systems [87].
- Reduced Peak Flows: Greenery-covered tall buildings slow down the rate of rainwater reaching the ground. The vegetation acts as a buffer, allowing rainwater to be absorbed, evaporated, or transpired before it enters drainage systems. This gradual release of water reduces peak flows during heavy rainfall events [89].
- Filtration and Purification: As rainwater moves through vegetation and soil, it undergoes natural filtration. Plant roots and soil particles capture pollutants and sediment, improving the quality of water that eventually reaches groundwater sources, rivers, and streams.
- Integration of Rainwater Management: Greenery-covered tall buildings can be designed with rainwater collection systems, such as rain gardens, green roofs, and vertical gardens. These systems capture rainwater for reuse in irrigation, reducing the demand for potable water for landscaping purposes [27].
- Enhanced Urban Resilience: In the face of climate change and increasing extreme weather events, greenery-covered tall buildings contribute to urban resilience by mitigating the impacts of heavy rainfall, reducing flood risks, and helping cities adapt to changing precipitation patterns [3,15].

Greenery-covered tall buildings can play a crucial role in rainwater management by retaining, absorbing, and gradually releasing rainwater through natural processes. These buildings contribute to reducing the strain on urban drainage systems, promoting sustainable water use, enhancing urban aesthetics, and improving the overall environmental health of cities.

### 5.1.9. Enhancing Aesthetics

Greenery-covered tall buildings can enhance aesthetics and improve the visual appeal of the buildings, their surroundings, and the overall urban landscape by introducing natural elements. Integrating plants, flowers, and green elements adds character and charm to the structure, attracting positive attention and potentially improving the imageability of a neighborhood [25–27,92]. The breakdown below elucidates this notion:

- Visual Integration: Lush vegetation on building façades, rooftops, and balconies softens tall buildings' harsh lines and rigid structures. This integration of greenery with architecture creates a harmonious and visually pleasing contrast [25].
- Natural Beauty: The vibrant colors, textures, and shapes of plants and flowers add a touch of nature's beauty to the otherwise concrete-dominated urban environment. Greenery-covered buildings create a sense of serenity and tranquility, offering a respite from the urban hustle and bustle [87].
- Seasonal Changes: The changing colors and blooms of plants throughout the seasons bring variety and dynamism to the cityscape. Residents and passersby can experience the joy of watching the transformation of greenery-covered tall buildings from spring blossoms to fall foliage [92].
- Enhanced Public Spaces: Greenery-covered tall buildings often integrate public spaces such as rooftop gardens, terraces, and plazas. These areas allow people to enjoy outdoor spaces with a natural backdrop, encouraging social interactions and relaxation [27].
- Event Venues: Similarly, rooftop gardens and terraces in greenery-covered tall buildings can serve as unique event venues. Hosting gatherings, parties, or corporate events in these spaces creates memorable experiences, reinforcing the building's image [87].

- Visual Identity: Greenery-covered buildings can become iconic landmarks that define the character of a city. They create memorable images and distinctive architectural features that residents and visitors associate with the urban landscape [27].
- Cultural Expression: Greenery-covered buildings can also incorporate elements of local culture and traditions, such as native plants and landscaping techniques. This adds a cultural layer to the aesthetics and reinforces the sense of place [92].
- Aesthetic Unity: Integrating greenery across a cluster of tall buildings creates a cohesive and harmonious urban environment. A skyline dotted with lush, verdant structures adds a unique and appealing character to the city [26].

Overall, greenery-covered tall buildings can enhance aesthetics by infusing urban spaces with nature's beauty, vitality, and serenity. They create visually appealing landmarks, provide opportunities for sensory engagement, and contribute to a more harmonious and inviting urban environment.

### 5.1.10. Branding and Marketing

Greenery-covered tall buildings can improve branding and marketing efforts for various stakeholders, including developers, businesses, and the city [38,40,62,93]. The following outlines these issues:

- Distinctive Identity: Greenery-covered tall buildings create a unique visual identity that differentiates them from traditional structures. This distinctiveness can become a recognizable brand element, making the building and its surroundings stand out in promotional materials and urban imagery [38].
- Innovative Image: Incorporating greenery into building design showcases innovation and forward thinking. Developers and businesses associated with such projects can position themselves as pioneers in sustainable and eco-friendly urban development. Businesses that occupy or develop such spaces can align their brand messaging with eco-conscious initiatives, attracting environmentally conscious consumers and investors [40].
- Civic Pride: Cities that embrace greenery-covered tall buildings communicate a commitment to improving the urban environment. Municipal authorities can use these projects to promote the city's livability, attracting residents, businesses, and tourists [68].
- Social Media Buzz: Greenery-covered buildings often become popular subjects on social media platforms. Their unique aesthetic draws attention, and users frequently share photos and experiences. This organic exposure can amplify the building's branding efforts [7].
- Storytelling: The process of designing and implementing greenery-covered tall buildings is an engaging story in itself. Brands can leverage this narrative to connect with audiences who appreciate the effort and creativity invested in the project [87].
- Tourist Attraction: Iconic greenery-covered structures can become landmarks that attract tourists. They provide photo opportunities and enhance the city's tourist appeal, contributing to positive reviews and word-of-mouth marketing [40].
- Investment Appeal: Greenery-covered buildings may attract interest from investors seeking projects aligned with sustainable and future-oriented urban development. This can enhance the financial viability of the project and attract capital [87].
- Differentiation: In competitive real estate markets, greenery-covered buildings differentiate themselves from standard offerings. This differentiation can lead to higher occupancy rates and premium rental or sales prices [40].
- Partnerships and Collaborations: The unique features of greenery-covered buildings can facilitate collaborations with complementary businesses, such as wellness centers, restaurants, or eco-friendly brands [93].

Greenery-covered tall buildings can offer a range of benefits for branding and marketing efforts. They provide a visually captivating identity, align with sustainability values, and create unique opportunities for engagement and promotion. Their innovative design

and positive impact on the urban environment make them compelling assets for effective branding strategies.

In summary, greenery-covered tall buildings may offer a wide array of potential benefits that span various aspects of urban living and development. From environmental improvements to social and economic advantages, these buildings can transform how we design and interact with our urban environments. Integrating greenery into tall buildings has the transformative power to reshape our urban landscapes and how we engage with our cities. This integration goes beyond architectural aesthetics; it promises to foster a more sustainable, livable, and harmonious urban environment. By bringing nature closer to where we live and work, we can reap many benefits that contribute to a better quality of life for current and future generations. From improved air quality and energy efficiency to enhanced community interactions and more vital branding, these greenery-covered structures represent a visionary approach to urban design that aligns with our evolving understanding of the importance of coexisting with nature in our built environments. Integrating nature into the vertical realm could create more sustainable, healthier, and aesthetically pleasing cities prioritizing human well-being and the planet's health. Table 4 summarizes the potential benefits of greenery-covered tall buildings.

**Table 4.** Summary of potential benefits of greenery-covered tall buildings.

| | | |
|---|---|---|
| 1 | Improved Air Quality | • Improve air quality by capturing and sequestering harmful pollutants like $CO_2$ and PM2.5, reducing their concentration in urban environments, especially in densely populated cities. |
| 2 | Urban Heat Island Mitigation | • Mitigate urban heat island effects by providing natural shade and cooling, blocking direct sunlight, absorbing heat, and using transpiration to absorb water vapor, reducing ambient temperatures, energy consumption, and greenhouse gas emissions. |
| 3 | Biodiversity Support | • Enhance biodiversity by creating new habitats for wildlife, promoting ecological balance, and fostering a healthier urban environment by providing food, shelter, and nesting sites. |
| 4 | Enhanced Well-Being | • Offer enhanced well-being, improving mental health and overall well-being.<br>• Provide a unique opportunity for occupants to connect with nature, reducing stress and enhancing relaxation.<br>• Enhance the quality of life, productivity, and creativity. |
| 5 | Energy Efficiency | • Offer energy efficiency through enhanced insulation, reducing reliance on heating and cooling systems and reducing carbon emissions. |
| 6 | Protection of Building Structure | • Protect the building envelope by shielding it from UV rays, regulating temperature fluctuations, and shielding it from harsh weather conditions.<br>• Help control moisture levels and reduce tension on the building's exterior surfaces. |
| 7 | Noise Reduction | • Reduce noise by absorbing and diffusing sound waves.<br>• Greenery acts as a barrier, preventing buildup and making indoor spaces quieter and more pleasant. |

**Table 4.** *Cont.*

| | | |
|---|---|---|
| 8 | Rainwater Management | • Improve urban water management by absorbing and storing rainwater through transpiration and evaporation.<br>• This reduces stormwater runoff, flooding risks, and pressure on drainage infrastructure.<br>• Additionally, greenery acts as a natural filter, improving water quality. |
| 9 | Enhancing Aesthetics | • Enhance aesthetics by transforming urban landscapes with unique façades, blending nature and architecture, and creating a harmonious blend of living, breathing spaces. |
| 10 | Branding and Marketing | • Elevate a city's reputation, showcase sustainable practices, attract media attention, and attract tourists, contributing to its eco-conscious identity and attracting talent, businesses, and investments. |

### 5.2. Challenges

However, amidst the positive aspects, less explored dimensions warrant attention: the potential disadvantages and challenges of integrating greenery into tall buildings, a focal point of this paper. The following section initiates a conversation that delves into concerns often associated with greenwashing. It raises vital questions surrounding construction costs, ongoing maintenance, and the long-term viability of greenery-covered tall buildings. Doing so paves the way for critical discussions and proposes potential remedies and solutions.

This discourse is a cornerstone for future empirical research, especially as data become increasingly available. By addressing these concerns head-on, we can ensure a well-rounded understanding of the complexities and tradeoffs associated with implementing greenery in high-rise architecture. This, in turn, lays the groundwork for a more comprehensive exploration of how greenery-covered tall buildings can genuinely contribute to sustainable and holistic urban development.

### 5.2.1. Construction Costs

Assessing the added construction costs for incorporating greenery into buildings is crucial in planning and decision making. This evaluation can help understand the financial implications of such designs and whether their benefits outweigh the initial investment [87]. Some key considerations when examining these added construction costs include:

- Materials and Infrastructure: Greenery-covered buildings require additional materials such as soil, containers, irrigation systems, and structural modifications to support the weight of the vegetation. These materials can contribute to increased construction costs. "In absolute terms, trees 100 cm in trunk diameter typically add 10 kg to 200 kg of aboveground dry mass each year (depending on species), averaging 103 kg annually. This is nearly three times the rate for trees of the same species at 50 cm in diameter and is the mass equivalent to adding an entirely new tree of 10–20 cm in diameter to the forest each year" [92], p. 15.
- Design and Engineering: Designing and engineering a building with greenery integration may involve specialized expertise and additional design iterations. Architects and engineers must collaborate to ensure the building can support the added weight and structural changes. Indeed, including trees within building structures demands meticulous structural planning, necessitating extra measures for reinforcement and strengthening. This is imperative to accommodate the trees' added weight and potential movement over time [38,62].

- Labor Costs: Installing green elements, such as planters, trellises, and irrigation systems, may require specialized labor. Skilled landscape installation and maintenance labor could also increase costs [62,64].
- Technology and Infrastructure: Integrating irrigation systems and technology to monitor and maintain the vegetation could require additional investments in infrastructure and equipment [68].
- Time is Money: Time carries economic implications, and integrating "innovative" features in buildings can extend construction timelines. The intricacies introduced by innovative designs often lead to more extended construction periods. For instance, the Bosco Verticale project in Milan spanned five years for completion, while a similarly sized non-greenery-integrated project in the same city typically requires around three years [60]. Consequently, the extension of construction duration translates to augmented expenses. To mitigate this, innovative approaches, specialized knowledge, and efficient techniques are crucial to streamline construction processes and reduce costs.

In the case of Bosco Verticale in Milan, to ensure trees' stability in the wind, they are tethered to the building using steel wires. The largest and most delicate trees were restrained in steel safety cages and secured in place once the superstructure was finished (Figure 9). According to the ARUP's structural engineers, "While all the medium and large trees have a safety cable to prevent the tree from falling in case the trunk breaks, the largest trees in those locations most exposed to wind have safety steel cages that restraint the root-bulbs and prevent them from overturning under major windstorms" [93].

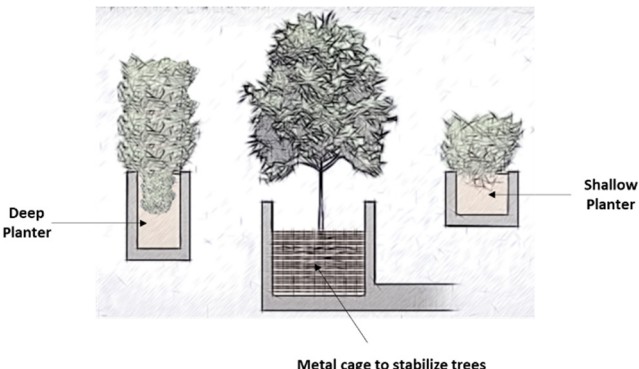

**Figure 9.** Planters in Bosco Verticale. (Sketch by author).

The "1000 Trees" project in Shanghai exemplifies these concerns, where researchers have questioned the cost-effectiveness of the employed vertical planters. The design and installation of a thousand vertical planters necessitate meticulous planning and engineering prowess. Guaranteeing adequate irrigation, efficient drainage, and optimal structural stability can present technical hurdles that might increase project costs [55]. Overall, integrating plant- and tree-covered elements into buildings necessitates a heightened structural potency compared to ordinary structures, resulting in increased construction costs.

### 5.2.2. Utility

Indoor versus Outdoor Space: An inherent challenge in integrating green elements into a building's façade is the delicate balance between maximizing outdoor space and maintaining indoor functionality. When incorporating green elements into a building's façade, architects and designers are tasked with harmonizing the desire to create lush, inviting outdoor spaces with the practical requirements of indoor functionality. This challenge emerges from the fact that both aspects are vital to a building's overall usability and the well-being of its occupants.

On one hand, maximizing outdoor space is essential for creating inviting environments that offer a sense of connection to nature, promote social interaction, and provide areas

for relaxation and leisure. Greenery-covered façades can extend these benefits upward, providing a vertical extension of green spaces that add aesthetic appeal, improve air quality, and contribute to the overall livability of the building and its surroundings. On the other hand, maintaining indoor functionality remains a crucial priority. Buildings are designed to offer safe, comfortable, and productive spaces for occupants to live, work, or engage in various activities. Balancing the integration of green elements with the need for proper insulation, natural light penetration, and adequate space utilization becomes a critical consideration. The challenge lies in ensuring that greenery does not compromise the quality of indoor environments by obstructing views, diminishing daylight, or causing issues related to moisture, humidity, and temperature regulation.

Furthermore, the arrangement and type of greenery must be thoughtfully chosen to enhance rather than hinder the building's core functions. For example, placing plants or trees near windows or entrances can provide shading, but careful planning is required to prevent obstruction of views or overgrowth that might require frequent maintenance. Addressing this challenge requires an integrated approach that involves collaboration between architects, landscape designers, and engineers. Incorporating advanced technologies, such as automated shading systems and responsive façades, can aid in managing the delicate balance between outdoor aesthetics and indoor functionality. Ultimately, achieving this equilibrium results in a greenery-enhanced building that seamlessly integrates nature into the urban environment while ensuring a comfortable and efficient interior space for its occupants.

The stepped garden design of ACROS Fukuoka creates terraces on the building's exterior, providing green spaces and promoting a biophilic environment. While this design has been celebrated for its innovation and architectural ingenuity, it also reduces the usable floor space for interior functions. Some critics have questioned whether this space could have been more efficiently utilized for additional offices, meeting rooms, or other functional areas, potentially maximizing the building's indoor space and functionality. Buildings in dense urban environments often face limitations on available land, making efficient use of space essential. Balancing the desire for greenery and outdoor spaces with the need for functional indoor areas can be complex for architects and developers [32,68].

- Balcony Use: The effectiveness of large balconies, particularly in regions with harsh weather conditions, has raised concerns among researchers and urban planners. In areas prone to extremely cold or hot weather, the practicality of these balconies can be seriously compromised, leading to reduced usage and essentially wasted space. Moreover, even on a single day, temperature fluctuations can be substantial, further limiting the usability of these outdoor spaces. Additionally, the wind tends to be more assertive at higher altitudes, making balconies on upper floors less comfortable for occupants [94].

Contrary to the idealized project renderings that often depict residents actively using balconies, a closer examination of real-world situations reveals that many tall buildings with balconies see minimal usage. Photographs of such buildings often show empty spaces, highlighting the disconnect between design intentions and user behavior. A quick internet image search can readily expose the extent of this issue, underscoring the challenges of creating genuinely attractive and functional balconies in all weather conditions. This discrepancy between design vision and user reality emphasizes the importance of taking a practical and context-aware approach when integrating balconies into high-rise buildings [57,94].

To address this challenge, architects and designers need to consider the unique climatic conditions of a location when planning and designing balconies. Thoughtful measures, such as incorporating weather-resistant materials, providing shading options, and designing flexible spaces that adapt to changing weather, can make balconies more versatile and appealing. Additionally, designers could explore innovative solutions that merge balconies with indoor spaces, creating a seamless transition between the two and maximizing their usability. In summary, the effectiveness of large balconies in unfriendly weather locations

is a multifaceted challenge that demands a holistic approach. By acknowledging the complexities of weather variations, wind patterns, and user preferences, urban planners and architects can create outdoor spaces that truly enhance the quality of life for residents, regardless of the weather conditions [94].

Along the same lines, tenant conversion of balconies into enclosed spaces is observed globally, particularly when interior living spaces are limited [94–96]. This practice can compromise the original design intent of the building, as these enclosed spaces may deviate from the architectural vision and disrupt the overall aesthetic [94–96]. Still, at the idea level, vegetation may occupy a sizable portion of balconies, raising questions about their cost–benefit effectiveness (Figure 10). Balconies are valuable real estate within a building, and dedicating a significant portion of them to vegetation can impact the overall utility of the space. Balconies with substantial greenery may require additional maintenance, irrigation systems, and structural considerations, all contributing to the overall cost. In the case of Bosco Verticale, researchers [97] have critiqued balconies for occupying a sizable portion of each floor plan (Figure 11). This has led to discussions about whether the architectural tradeoff, which emphasizes greenery on the façade, might inadvertently hinder occupants' practical use of these outdoor spaces.

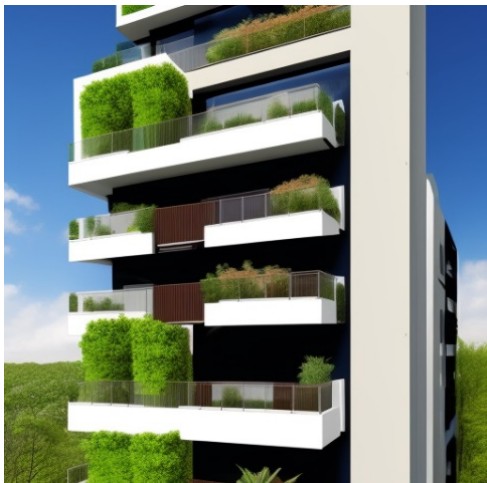

**Figure 10.** Integrating vegetation may occupy valuable and limited space on balconies. (Sketch by author).

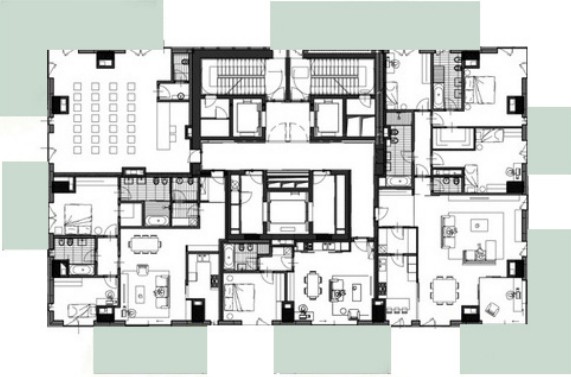

**Figure 11.** Balconies occupy a sizable portion of every floor plan in Bosco Verticale. (Sketch by author, adapted from [22]).

Therefore, balancing the desire for greenery with the functional needs of residents requires a thoughtful approach. Architects and designers should consider the diverse ways occupants might utilize their balconies and aim for a harmonious integration of vegetation and human activity. Solutions could involve designing multifunctional balcony spaces that

accommodate both green elements and occupant needs, thereby maximizing the benefits of these outdoor areas. Ultimately, the challenge lies in finding the proper equilibrium between incorporating vegetation, maintaining usability, and achieving the original design goals of the building. By carefully addressing these considerations, designers can ensure that balconies remain both attractive and functional components of high-rise buildings [94,95,97].

- Terrace Use: Similarly, research has examined Singapore's Oasia Hotel, which has come under scrutiny for allocating 40% of its volume to green, open-air terraces. While these green spaces align with the hotel's biophilic design approach and offer various environmental and social advantages, a critical tradeoff exists between the extent of greenery and the hotel's revenue-generating capacity [98]. The decision to dedicate a significant portion of the building's volume to green terraces has implications for the total number of hotel rooms that can be accommodated. Consequently, the hotel's capacity to generate revenue from room rentals may be diminished due to reduced available guest accommodations. This tradeoff prompts a delicate balance between the desire to incorporate extensive green elements and optimizing the hotel's financial performance, particularly regarding return on investment (ROI).

While green spaces within the hotel contribute to a sense of connection with nature and provide wellness benefits for occupants, this must be weighed against potential impacts on the hotel's operational efficiency. By reducing the total number of available rooms, there is a potential to affect the hotel's occupancy rates. Fewer rooms mean the hotel can accommodate fewer guests simultaneously, potentially affecting its ability to generate consistent revenue. Consequently, allocating a substantial volume to green terraces may impact the hotel's overall revenue potential and financial viability. This highlights the intricate challenge faced by designers and developers when seeking to achieve a balance between environmental sustainability, guest experience enhancement, and economic success.

Considering these issues, design strategies that maximize greenery while minimizing the impact on revenue potential must be explored. Creative approaches might involve optimizing the placement and configuration of green spaces to enhance the building's overall appeal while mitigating the reduction in room availability. This underscores the need for a holistic and multidisciplinary approach to architectural design that integrates diverse considerations to create vibrant, sustainable, and economically viable built environments [98].

- Sunlight: Incorporating expansive vegetated balconies and terraces adorned with abundant plants and substantial trees presents a potential challenge concerning obstructing sunlight and natural daylight from entering indoor spaces. This can result in a heightened reliance on artificial lighting, leading to increased electricity consumption and subsequent utility bills. The situation is further compounded by the interplay between shadows cast by vegetation and cantilevered architectural elements such as balconies and terraces.

The combination of these factors contributes to the casting of substantial shadows, exacerbating the reduction of sunlight penetration and creating an atmosphere of gloom within indoor spaces. This effect is particularly pronounced when large, cantilevered balconies or terraces are heavily vegetated. Such configurations effectively impede the flow of natural light and sunrays into interior spaces. For instance, in the Bosco Verticale project, balconies are designed to cantilever 3.5 m with a thickness of 28 cm, intensifying the impact of this challenge (Figure 12).

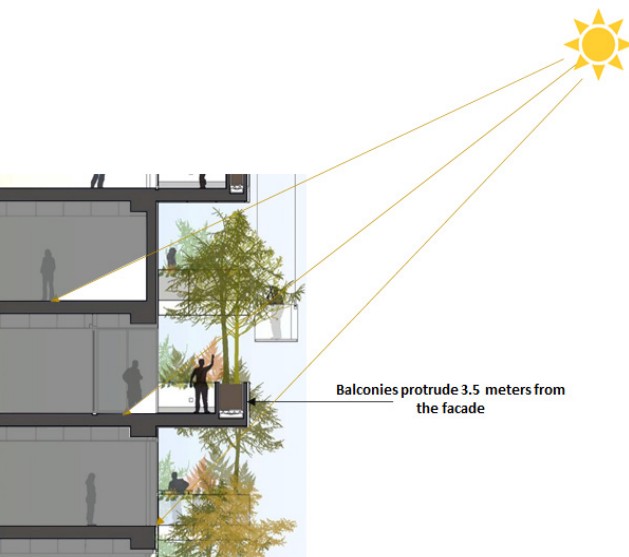

**Figure 12.** Large, cantilevered balconies, lush greenery, and tall trees block sunrays and natural daylight from Bosco Verticale's indoor spaces. (Sketch by author, adapted from [22]).

Similarly, the ACROS Building exhibits a similar concern. The presence of plants and trees obscures the entirety of the southern façade from direct sunlight, leading to dimly lit indoor areas. Furthermore, this vegetation cover hinders natural ventilation, potentially affecting the overall indoor air quality and healthiness of the spaces (Figure 13).

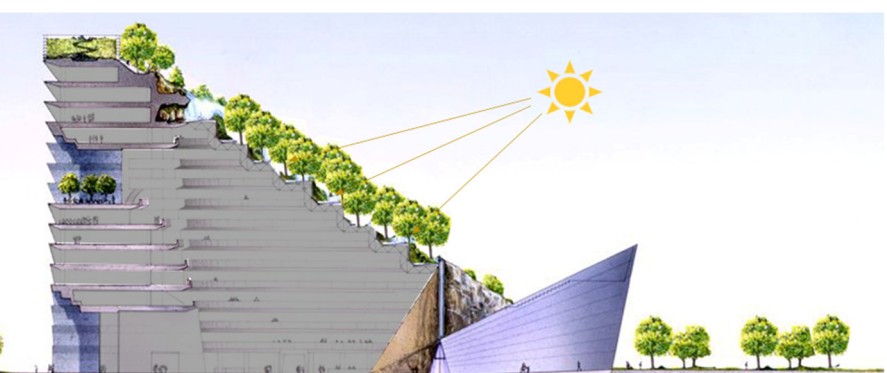

**Figure 13.** The vegetated terraces block sunrays and natural daylight from entering ACROS's indoor spaces. (Sketch by author, adapted from [32]).

Recognizing the significance of sunlight exposure in fostering human well-being is essential. Sunlight exposure facilitates the synthesis of vitamin D in the epidermis, contributes to blood pressure regulation and appetite suppression, and offers protection against obesity, type 2 diabetes, and certain autoimmune disorders. Consequently, the obstruction of sunlight due to excessive vegetation could have implications for the health and well-being of occupants [72].

This concern becomes increasingly relevant as incorporating vegetated balconies gains traction in greenery-covered tall buildings worldwide (see Table 2) [29]. Many such buildings have blindly embraced this architectural feature across diverse climatic contexts, often without considering the associated challenges. For instance, the viability of extensive vegetation in extreme climates, such as the scorching conditions of Dubai or the cold weather of the Netherlands, raises questions about the practicality and sustainability of balconies and terraces in these environments.

Therefore, integrating large, vegetated balconies and terraces presents a potential downside due to their capacity to obstruct natural light and impact indoor spaces. As more

buildings adopt this concept, addressing the architectural, climatic, and health-related implications is imperative to ensure that greenery-covered tall buildings genuinely deliver on their promise of holistic sustainability and occupant well-being.

- Building's Façades and Vegetation: The amount of greenery in these buildings' different façades seems equal regardless of their solar orientation. The southern, northern, western, and eastern façades have similar amounts of vegetation, where light and solar conditions differ. Green design teaches that each façade should receive different treatment to adequately address wind direction and solar orientation. Eastern and western façades may need vertical fins to protect from sunrays, southern façades may need light shelves, while northern façades (which do not receive sunrays) need none.
- A similar problem prevails in Oasia Downtown in Singapore. The building's four façades receive the same architectural and planting treatment, violating sustainable design principles. By providing the same treatment on all four façades, Oasia Hotel Downtown may miss out on opportunities to maximize the benefits of greenery integration. Different orientations and environmental conditions on each side of the building could be better addressed through tailored planting schemes and design strategies. For example, particular façades may require more shade-providing vegetation to reduce heat gain, while others could benefit from plant species that enhance wind protection and ventilation [45,98].
- Carbon Emissions: Incorporating trees and vegetation into high-rise buildings introduces a range of factors that must be weighed, including their potential environmental costs. One key concern is the carbon emissions associated with the production of materials used in construction, particularly steel and concrete. These emissions contribute to the broader carbon footprint issue, which significantly impacts sustainable urban development.

The Bosco Verticale project provides a pertinent example of this challenge. A study estimated that the concrete production employed in constructing the towers' balconies resulted in approximately 990 tons of $CO_2$ emission. In contrast, the trees and plants integrated into the building's design can sequester around 18 tons of $CO_2$ annually. This presents a scenario where the tower's greenery could take around 55 years to offset the carbon emissions produced solely by constructing its balconies [67].

This discrepancy highlights the complexity of the carbon emissions equation associated with greenery-covered tall buildings. While the integrated vegetation offers carbon sequestration benefits over time, the initial carbon emissions generated from construction activities, mainly concrete production, present a substantial upfront cost regarding environmental impact. This underscores the need to take a comprehensive view of sustainability when evaluating the environmental benefits of such projects.

In essence, the carbon emissions attributed to the production of building materials counterbalance the long-term carbon sequestration potential of greenery-covered tall buildings. This dynamic necessitates a holistic assessment of the ecological impact of these buildings, considering both their initial environmental costs and their capacity to contribute positively to carbon sequestration and reduction over time. As the pursuit of sustainable urban development continues, finding ways to minimize carbon emissions during construction while maximizing the long-term environmental benefits of greenery integration becomes a critical consideration [67].

### 5.2.3. Fire

Integrating abundant organic materials on building façades in the form of vegetation can introduce a unique challenge: the risk of fire. This novel building design approach prompts careful consideration of fire safety measures to protect both occupants and the building itself. National building codes often mandate that designers and engineers devise solutions that prevent fire spread through exterior cladding or façades [91].

In response to this concern, specific precautions must be taken to manage the fire risk posed by the presence of vegetation. Compliance with relevant regulations is paramount,

necessitating strategies that mitigate potential fire hazards. These strategies encompass various aspects, including the maintenance and care of the vegetation itself.

One crucial aspect of managing fire risk involves controlling the size and condition of the vegetation. Keeping the vegetation within specific size limits and preventing it from becoming excessively dry are essential measures to reduce the potential for fire ignition and spread. Additionally, incorporating appropriate fire suppression and evacuation systems is essential to ensure swift response and safe evacuation during a fire emergency.

Adequate care and maintenance of vegetation are critical components of fire prevention. To minimize the risk of fire, maintaining a low percentage of organic matter in the soil can help prevent ignition. Regular pruning and irrigation routines also play a role in reducing the likelihood of fire outbreaks [91].

In summary, incorporating organic materials in the form of vegetation on building façades introduces fire safety considerations that must be carefully addressed. Adhering to building codes and regulations, implementing proper fire prevention measures, and ensuring consistent vegetation maintenance are critical steps in managing the fire risk associated with these innovative greenery-covered building designs. By doing so, architects and developers can balance sustainable design and occupant safety.

### 5.2.4. Building Codes

- Hurdles: Innovative projects face passing building code requirements. Design justification is needed since they have not been built before, and developers and architects need the authorities' backing. For example, some claim Bosco Verticale received the government's support because it was built before the Milan Expo [67]. Therefore, the local authorities supported the project as a showcase project for the city, which does not always happen. It is another game for cities with strict building codes, such as New York City or a historic district that advocates preserving the historic fabric and prohibits "outlier" buildings, such as buildings with very different perceptual characteristics [91].
- Incentives: Singapore is an excellent case illustrating the government's support for integrating greenery in buildings, including tall ones. The Skyrise Greenery Incentive Scheme (SGIS) provides funding of up to fifty percent of installation costs for rooftop and vertical greenery initiatives on existing buildings, with a cap of USD 200 per square meter for rooftop greenery and USD 500 per square meter for vertical greenery. NParks instituted the SGIS in 2009 to increase Singapore's greenery provision [99]. Projects supported by the SGIS include edible community rooftop gardens, recreational and therapeutic rooftop gardens, extensive green roofs, and luxuriant verdant green walls on more than 200 buildings. Some project examples include Oasia Hotel Downtown, Khoo Teck Puat Hospital, the National Parks Board (NParks) headquarters, and the School of the Arts. With these various initiatives and support from developers, building professionals, and owners, Singapore is well into achieving a target of 200 hectares of sky greenery by 2030 and towards a greener biophilic Singapore [99,100].

### 5.2.5. Maintenance

The ongoing upkeep required by tall buildings covered in vegetation is a crucial element that affects their long-term functionality and success. Unfavorable outcomes have been caused by some examples of this architectural typology that have received little care. For instance, certain high-rises in Chengdu, China, covered in plants and trees, revealed a lack of upkeep and attention to their landscaping. As a result, these buildings suffered from a lack of suitable patronage. Many people refused to occupy them or abandoned them. In the instance of Oasia Downtown, Singapore, the enormous vertical garden that covers the building's green façade needs upkeep. For the hotel guests, the maintenance operations make some noise and cause disturbance. The ACROS building's rooftop garden has drawn criticism for needing continuous watering, pruning, and pest management [71].

Let us delve deeper into the challenges and considerations related to maintenance:

- Integrity and Aesthetic Concerns: Living and perpetually growing vegetation can challenge the façade's integrity and aesthetics. Without proper maintenance, plants and trees may grow uncontrollably, covering windows and blocking natural light from entering indoor spaces. This growth can lead to a lack of visibility and create an unappealing visual appearance for the building [93].
- Pruning and Trimming: Regular pruning and trimming of vegetation are necessary to prevent overgrowth and maintain a desired appearance. Failure to trim trees and plants can lead to branches and leaves encroaching on windows and façades, hindering occupants' views and impacting the overall aesthetics of the building [62].
- Impact on Indoor Spaces: Unchecked vegetation growth can directly impact indoor spaces' quality. Plants that cover windows prevent natural light from entering, leading to darker and less inviting interiors. This situation can necessitate increased reliance on artificial lighting, contributing to higher energy consumption [38].
- Risk of Pests and Invasive Species: Neglected vegetation can become a breeding ground for pests and potentially invasive species. Lack of maintenance and pruning can create hiding spots for pests and insects, leading to infestations that can quickly spread throughout the building. This poses health risks to occupants and undermines the overall appeal of the building [93].
- Specialized Expertise: Maintaining greenery-covered tall buildings requires specialized horticulture and landscape management expertise. Gardeners must possess knowledge of the specific plant species used, their growth patterns, irrigation requirements, and pest control strategies. Richard Hassell explained that this building typology involves a team of flying gardeners who are both "Spidermen" and experts on local vegetation. He referred to the Vertical Forest example, where they fly around the building every four months. They hang by rope from the roof's edge, descend, and jump between balconies [45]. As mentioned earlier, the Vertical Forest has exploded with wildlife since its construction, providing a habitat for over 1600 birds and butterfly species. However, that in itself creates a maintenance issue. The plants may become unhealthy without proper care, affecting the building's overall appearance [93].
- Watering and Irrigation: Regular watering and irrigation are essential to keep plants and trees healthy. However, inadequate watering can lead to plant stress and wilting, while overwatering can result in water pooling, leaks, and potential damage to the building structure. Careful monitoring of irrigation systems is necessary to maintain a balance [93].
- Structural Considerations: As greenery-covered buildings age, structural wear and tear may occur. Maintenance should include regular inspections to identify water leaks, cracks, and building structure weaknesses. Notably, most of the completed towers of the plant- and tree-covered prototypes are young and have not passed the test of time. The wear and tear effect may create problems in the irrigation system, such as water leaks, which could be a serious problem in high-rise buildings since water leaks affect multiple floors below. Water can seep into walls, ceilings, and floors, causing damage to the building's structure and interior finishes. It can also lead to the growth of mold and mildew, posing health risks to occupants and further compromising the building's integrity [101].

Several measures can be taken to mitigate the risk of water leaks in greenery-covered tall buildings. Regular inspections of the irrigation system and building envelope are crucial to detect any signs of leaks or potential issues. Early detection allows for timely repairs and prevents further damage. High-quality irrigation and waterproofing materials are essential to ensure durability and longevity. Investing in robust, reliable components can help prevent water leaks and reduce maintenance needs.

- Root Growth and Structural Damage: Similarly, the roots of trees and plants may go out of control over time and cause cracks in the building's structure and façades. Tree roots have the potential to exert considerable force as they grow, seeking out moisture and nutrients from the soil. If tree roots encounter weaknesses in a building's

foundation or cracks in the walls or façades, they can exploit these vulnerabilities and exacerbate the damage. As the roots grow and expand, they can push against the building's structure, leading to cracks and potential instability. To address this challenge, architects, engineers, and landscape designers must carefully plan and implement greenery-covered building projects. Proper selection of tree and plant species with non-invasive root systems can help mitigate the risk of damage. Additionally, using specially designed root barriers and structural systems can help guide root growth away from critical areas of the building [93].

- Environmental Considerations: To create a healthy indoor environment, addressing potential mold growth requires proper ventilation and humidity control. The building should prevent harmful mold. Under some circumstances, certain types of molds, such as Stachybotrys and Aspergillus, can produce poisons known as mycotoxins [102]. Severe sickness could occur because of mycotoxin exposure. Therefore, the building's façade should be permanently sealed to prevent undesirable molds and insects from crawling into interior spaces. Any maintenance carried out at higher elevations will be more expensive and complicated. Adequate ventilation is crucial to control indoor humidity levels. High humidity can create a conducive environment for mold growth. Proper ventilation helps to expel excess moisture and maintain a dry indoor environment [87,93].

Therefore, the continuous maintenance demanded by greenery-covered tall buildings is a multifaceted challenge that involves balancing aesthetic appeal, structural integrity, indoor comfort, and ecological health. Effective maintenance practices, specialized expertise, and a commitment to addressing potential issues are vital to ensuring the success and sustainability of these innovative architectural concepts.

### 5.2.6. Watering Costs

Sustaining a vertical forest in an urban environment demands significant water resources for irrigation. Critics have raised concerns about the amount of water required to keep the vegetation healthy and whether this level of water consumption is sustainable in water-stressed cities such as Milan. Water scarcity is a critical issue in many urban areas around the world where the availability of freshwater resources may be limited, especially during periods of drought. The large-scale irrigation required to maintain a vertical forest can strain the city's water supply and exacerbate existing water challenges [87].

Therefore, depending on the water requirement for different plants, availability of rainwater, and local fees, water costs could be high, making this prototype unaffordable to some segments of society. For example, in tropical regions (as with Singapore), growing trees and plants are relatively effortless due to supportive climatic conditions that feature an abundance of rainwater and humidity. With a naturally high level of rainfall, the need for additional water resources for irrigation is minimized, resulting in lower water consumption and reduced strain on local water supplies. The self-sustaining nature of the tropical climate in providing water and humidity can lead to lower watering costs for greenery-covered buildings than locations with drier climates.

To reduce water consumption, architects should choose indigenous plants that require minimal watering. Opting for drought-resistant plant species requiring less water can reduce irrigation needs while maintaining a lush green appearance. Shading could be crucial to cut watering costs. Employing shading devices and elements could help reduce watering costs and improve the health of plants. Implementing advanced irrigation technologies, such as drip irrigation or intelligent irrigation systems, can minimize water wastage by delivering water directly to the plant roots where it is most needed. Similarly, capturing and storing rainwater for irrigation can offset the demand for potable water and reduce the strain on the city's water supply. Utilizing treated greywater from showers, sinks, and other non-potable sources can be an environmentally friendly way to water the vertical forest. Also, solar-powered irrigation systems should be examined since they may offer a nature-based solution (NBS) for sustainable water management [87].

5.2.7. Plants' Health

- Illness: Disease can affect any plant, whether wild or cultivated. When infected by a disease, plants can become ill, just like humans. Plant disease is described as the state of improper local or systemic physiological functioning of a plant resulting from the continuous, sustained "irritation" generated by phytopathogenic organisms. There is a wide range of plant-infecting bacteria, fungi, viruses, and nematodes. Some infections infest the roots from below ground, while others thrive in the air and attack the plant's leaves [103]. Pathogens affecting plants, and the variants of those pathogens that have arisen over time, are a constant source of discoveries for plant pathologists. Plant healthcare gives plants the essential nutrients to flourish while safeguarding them from insects and illness [104]. The treatments incorporate fertilization, insect control, and disease prevention. It is necessary to treat destructive plant pests when they are active each year to contribute to population reduction and maintain the health of plants. If we do not take this precaution, there is a good chance that the plant's existence will be cut short by illnesses or pests that are harmful to it. As such, sustaining the health of plants entails added burden and cost [105].
- Meteorological Conditions: Indeed, the health and well-being of trees and plants in tall structures clothed in greenery can be significantly impacted by weather conditions at higher altitudes. The particular difficulties that higher locations provide can have various effects on vegetation.

Temperature Extremes: As buildings rise to higher altitudes, they are more exposed to temperature fluctuations and extremes. Trees and plants in these environments may experience colder temperatures, stronger winds, and increased exposure to harsh weather conditions. Frost, snow, and ice accumulation can damage leaves, branches, and even the root systems of plants, leading to stress and decreased overall health [87].

Wind Exposure: Also, tall buildings are often subjected to stronger winds at elevated heights. Wind can desiccate plants by causing rapid moisture loss through transpiration. This can lead to water stress, wilting, and reduced growth [87].

Sunlight Intensity: Strong winds can physically damage delicate foliage and disrupt the growth patterns of trees and plants. Further, reduced atmospheric filtering may expose trees and plants to more intense sunlight at higher altitudes. While sunlight is essential for photosynthesis, excessive sunlight can lead to sunburn, leaf scorching, and dehydration. These conditions can weaken plants and hinder their ability to thrive [15].

- Roots' Growth. The lack of space for root growth could impact the health of large trees. The limited available space for root growth is a critical factor that can significantly impact the health and growth of large trees in urban environments, including within greenery-covered tall buildings. Tree roots play a vital role in providing stability, absorbing water and nutrients, and supporting the overall health of trees [93]. When trees are planted in environments with restricted root space, several challenges can arise:

Stunted Growth: Trees require sufficient space for their roots to spread and establish a stable foundation. Limited space can result in stunted root growth, affecting the tree's overall growth and development. Without proper root expansion, trees may struggle to reach their full potential in height, canopy size, and foliage density [77].

Reduced Nutrient and Water Uptake: Roots absorb water and nutrients from the soil. The tree's access to essential resources becomes compromised when root space is constrained. This can lead to water stress, nutrient deficiencies, and poor overall health. Inadequate water and nutrient uptake can result in thinning foliage, discoloration, and increased vulnerability to pests and diseases [87].

Soil Compaction: In urban environments, soil compaction is a common issue due to construction activities, foot traffic, and the weight of buildings. Compacted soil restricts the movement of air, water, and nutrients, making it even more challenging for tree roots

to penetrate and thrive. Compacted soil conditions can suffocate roots and hinder their ability to grow and function effectively [93].

-   Increased Risk of Diseases and Pests: Crowded root systems and stressed trees are more susceptible to diseases and pest infestations. When roots are confined to a small area, they are more likely to compete for resources and become vulnerable to pathogens and pests that thrive in compromised environments [10].

To address the challenge of limited root space and support the health of large trees in greenery-covered tall buildings, several strategies can be considered:

Soil Volume Enhancement: Providing adequate soil volume for root growth is essential. Techniques like root cell systems and structural soil systems can create spaces for roots to grow without compromising building foundations [66].

Species Selection: Choosing tree species with less aggressive root systems or those that are naturally suited to confined spaces can help mitigate the effects of limited root growth [27].

Tree Pit Design: Designing tree pits that allow for vertical root growth, aeration, and moisture retention can enhance root development [14].

Soil Aeration: Implementing soil aeration techniques, such as radial trenching or vertical mulching, can alleviate soil compaction and encourage root growth [93].

Regular Monitoring and Maintenance: Regular monitoring of tree health, soil conditions, and root growth is essential. Proper maintenance practices, including pruning, fertilization, and irrigation, can help mitigate the impact of limited root space [93].

Addressing the challenge of limited root space is crucial for successfully integrating large trees in greenery-covered tall buildings. By prioritizing the health and well-being of tree roots, architects, landscape designers, and urban planners can contribute to the long-term vitality and sustainability of urban green spaces.

-   Light/Shade Balance: Establishing light/shade balance becomes an issue of concern. Overall, selecting the right plants for each façade and elevation is important. The French botanist Patrick Blanc made a great effort in choosing the plants for Sydney's One Central Park tower so that they would thrive in the city's unique environment and seasonal changes. Vegetation is successfully adapted to its location and growth conditions by employing acacias (wattles) and Poa (grasses) on higher levels and Goodenia (hop bush) and Viola (native violet) on lower levels. Over 1100 square meters of wall space are home to a wide variety of plants, most local to Sydney [38,106]. So far, these plants have shown resilience in withstanding hot, dry, and windy Australian summers.

### 5.2.8. Experiencing Nature

-   "Compromised" Experience: Innovative methods to bring nature into sky living are appreciated. However, they may not offer tenants the whole experience of interacting with nature. For example, planting in greenery-covered high-rises relies on artificial watering systems, depriving residents of the natural experience of watering plants, checking on their needs for water, and observing the effects of watering them on their growth and well-being. Watering plants activates interest in weather conditions (sunny, cloudy, rainy, etc.) and awareness of solar orientation and sun path [107].

Similarly, professional gardeners carry out pruning, obviating residents' interactive experience with plants and trees. Checking on plants' health is a caring human experience. Residents may want to check plants' growth, needs, and soil conditions. Similarly, they may enjoy "digging" and planting their own vegetation. Tenants may want to perform seeding, transplanting, pruning, and harvesting. These tasks and aspects of interacting with vegetation can positively affect human well-being, cognition, and psychology [108]. Further, plants seem to be given in these buildings, and tenants have no say in choosing the ones they love and desire.

-　　Acrophobia: Some people have acrophobia, the fear of heights that can cause significant distress and impairment. People with acrophobia may experience panic attacks, nausea, dizziness, sweating, trembling, and difficulty breathing when exposed to high places or situations involving height. Consequently, they may not feel comfortable interacting with nature in vegetated balconies and terraces on the upper floors. Interacting with nature involves engaging all our senses, being present in the moment, and feeling awe and gratitude for the natural world [109]. As such, this solution has some limitations and challenges to achieving biophilic design by not offering a fully immersive experience with nature [110].

5.2.9. Exclusivity and Gentrification

-　　Affordability and Socioeconomic Diversity: The issue of cost-effectiveness is a crucial consideration when it comes to implementing expansive architectural designs that include substantial vegetal components. The elevated construction and maintenance costs associated with these innovative structures are often transferred to property owners or renters, which can result in reduced accessibility for a significant segment of the population. Incorporating extensive greenery, vertical gardens, and sustainable features can escalate upfront expenses during construction and ongoing maintenance [87].

Consequently, the cost of residential units or rental rates in such buildings may be elevated, potentially making them less attainable for low-income individuals and families. This economic barrier raises questions about these developments' inclusivity and socioeconomic diversity. If only a specific segment of the population can afford to live in these buildings, it might limit the diversity of people who can enjoy the benefits they offer.

-　　Gentrification: Furthermore, the "luxury" nature of some of these developments, which often emphasize unique design features and premium amenities, can contribute to gentrification in the surrounding area. As these greenery-covered tall buildings become sought-after symbols of high-end living, they may attract wealthier residents and investors. The influx of affluent individuals can increase demand for nearby properties, potentially driving up property prices and rents in the neighborhood. Some critics have pointed out that the luxury nature of One Central Park in Sydney and Le Nouvel KLCC in Kuala Lumpur could contribute to gentrification in the surrounding area, potentially leading to increased property prices and displacement of lower-income residents [73,75]. Likewise, others have argued that Bosco Verticale's iconic design might have influenced nearby property values, potentially leading to gentrification and displacement of lower-income residents.

Gentrification can have both positive and negative impacts on a community. On the one hand, it may lead to improved infrastructure, amenities, and investment in the area. However, on the other hand, it can also result in the displacement of lower-income residents who may be unable to afford the rising living costs. This can lead to issues of social inequality and loss of community cohesion.

To address these concerns, urban planners and developers must carefully consider the social impact of greenery-covered tall buildings. Implementing policies that promote affordable housing and mixed-income developments can help maintain diversity and inclusivity in urban areas. Additionally, encouraging developers to incorporate affordable housing components within luxury developments or allocating space for public amenities can contribute to a more balanced and sustainable urban environment.

By striking a balance between promoting innovative and sustainable architecture while ensuring affordability and inclusivity, cities can harness the benefits of greenery-covered tall buildings without exacerbating issues of gentrification and social inequality. Collaborative efforts between developers, local governments, and community stakeholders are crucial in creating vibrant and accessible urban spaces for everyone.

### 5.2.10. Urban Wildlife and Pest Control

Greenery-covered buildings in urban environments can attract various urban wildlife and pests, presenting a dual challenge. While these buildings enhance biodiversity by providing habitats for beneficial species like birds and insects, they can also introduce difficulties in managing unwanted wildlife and pests. Some of the critters that might be drawn to the vegetation, such as insects, rodents, and snakes, can create an unsuitable living environment for humans. Even seemingly minor issues, such as the presence of mosquito infestations, can significantly deter individuals from choosing to live in high-rise buildings [71]. Therefore, developers and building owners must implement effective maintenance measures to address these issues.

- Greenery Choices: To counteract potential pest problems, developers might opt for landscaping and greenery choices less prone to attracting pests. This can reduce the risk of unwelcome infestations. Implementing an integrated pest management (IPM) strategy involving a combination of pest control techniques can be helpful. These techniques may include biological controls, physical barriers, and targeted use of pesticides only as a last resort [93].
- Inspections: Regular inspections and maintenance of the greenery are essential for early detection and resolution of pest issues. Taking prompt action can prevent infestations from spreading. Non-harmful deterrents, such as reflective surfaces, noise-emitting devices, or netting, can discourage birds and other wildlife from nesting in undesirable areas [65].
- Waste Disposal: Proper waste disposal and minimizing potential food sources for pests, such as rodents and insects, are essential preventive measures. By considering these aspects, building owners and developers can balance the benefits of greenery and the need to manage potential pest challenges, ensuring that these high-rise structures remain appealing and habitable for their human occupants [73].

This section examined the various obstacles associated with incorporating vegetation in high-rise structures, with particular emphasis on the financial implications during the construction phase, the continuous upkeep requirements, and the sustainability of such initiatives in the long run. Construction costs encompass several factors, such as the augmented weight of trees, installation of irrigation systems, and ongoing upkeep. The latter necessitates meticulous planning and the application of engineering knowledge. Achieving a harmonious integration of environmentally friendly components and the practical requirements of inhabitants presents a multifaceted undertaking, necessitating designers to account for the distinct climatic circumstances of a given area, integrate resilient materials capable of withstanding adverse weather conditions, offer shading alternatives, and conceive adaptable spaces capable of accommodating fluctuations in weather patterns. Incorporating vegetation in tall buildings gives rise to apprehensions over ecological impacts and carbon footprints and necessitates the implementation of fire safety protocols. The Skyrise Greenery Incentive Scheme in Singapore provides help for the integration of greenery in buildings; nonetheless, the issue of upkeep is of utmost importance. Their significant water use may compromise the sustainability of implementing large-scale vertical forest designs in water-stressed urban areas. Table 5 summarizes the potential drawbacks and challenges of greenery-covered tall buildings.

**Table 5.** Summary of potential drawbacks and challenges of greenery-covered tall buildings.

| | | |
|---|---|---|
| 1 | Construction Costs | • Construction costs for plant- and tree-covered buildings, irrigation systems, and maintenance should be considered.<br>• Trees add weight and size, requiring special consideration and strengthening.<br>• Vertical planters require careful planning and engineering expertise.<br>• Innovative buildings require additional construction time, which can increase costs.<br>• Horizontal landscaping costs are often lower than vertical landscaping costs, and innovative models and techniques are needed to reduce construction time. |
| 2 | Utility | • Integrating green elements into buildings faces challenges, such as the tradeoff between outdoor and indoor space.<br>• Large balconies, particularly in unfriendly weather locations, can waste space and may not be used effectively.<br>• The use of vegetation may also occupy a sizable portion of balconies, raising questions about their cost–benefit effectiveness.<br>• Vegetated balconies and terraces can block sunlight and natural daylight, increasing electricity bills and reducing indoor space health.<br>• The amount of greenery in buildings' façades should be tailored to different orientations and environmental conditions.<br>• Additionally, high-rises may have environmental costs to accommodate trees, such as carbon emissions from steel and concrete production.<br>• Balancing the desire for greenery and outdoor spaces with the need for functional indoor areas can be complex for architects and developers. |
| 3 | Fire | • The novel design poses fire risks due to abundant organic materials on façades, necessitating strict building codes, fire suppression systems, and proper maintenance. |
| 4 | Building Codes | • Innovative projects often face passing building code requirements, requiring design justification and authority support.<br>• Singapore's Skyrise Greenery Incentive Scheme (SGIS) provides funding for rooftop and vertical greenery initiatives on over 200 buildings, aiming to achieve 200 hectares of sky greenery by 2030. |
| 5 | Maintenance | • The Bosco Verticale building prototype faces challenges in maintaining its façade due to the continuous growth of plants and trees.<br>• The vertical garden requires specialized gardeners, such as flying gardeners, to ensure the building's integrity.<br>• The lack of maintenance can lead to issues like water leaks, damaging the structure and interior finishes.<br>• To mitigate these risks, regular inspections, high-quality irrigation, and investing in reliable components can help prevent water leaks and reduce maintenance needs.<br>• Architects should plan greenery-covered projects, use non-invasive root systems, and use root barriers.<br>• Buildings should prevent mold and insects and maintain proper ventilation to control indoor humidity levels. |
| 6 | Watering | • Water scarcity in urban areas can strain water supplies, making vertical forests unaffordable for some.<br>• To reduce water consumption, architects should choose drought-resistant plants, use shading devices, implement advanced irrigation technologies, capture rainwater, use treated greywater, and explore solar-powered irrigation systems.<br>• Tropical regions with abundant rainfall can also offer lower watering costs. |
| 7 | Plants' Health | • Plant disease affects plants, causing improper physiological functioning due to continuous irrigation by phytopathogenic organisms.<br>• Plant healthcare includes fertilization, insect control, and disease prevention.<br>• Maintaining plant health requires additional resources and costs.<br>• Plants' health is affected by weather conditions and light/shade balance.<br>• French botanist Patrick Blanc adapted plants for Sydney's One Central Park tower, using local plants for resilience in various conditions. |

**Table 5.** *Cont.*

| 8 | Experiencing Nature | • Innovative methods for incorporating nature into sky living spaces may not fully offer residents the whole experience of interacting with nature.<br>• Artificial watering systems and professional gardeners may deprive residents of the natural experience of watering plants, pruning, and planting their own vegetation. |
|---|---|---|
| 9 | Exclusivity and Gentrification | • Greenery-covered tall buildings can be less accessible to lower-income individuals due to higher construction and maintenance costs.<br>• The luxury of these buildings can attract wealthier residents, increasing property prices and displacement. |
| 10 | Urban Wildlife and Pest Control | • Urban greenery attracts wildlife and pests, making buildings potentially unfit for human habitation.<br>• Developers should implement an IPM approach, use non-harmful deterrents, and ensure proper waste disposal.<br>• Regular inspections, deterrents, and waste reduction can help prevent infestations and maintain biodiversity. |

## 6. Conclusions

This study thoroughly examines an emerging architectural trend and its capacity to shape urban landscapes profoundly. It addresses a broad audience, including architects, developers, scholars, and the public. The overarching goal is to create urban habitats that strike an ecological equilibrium, teem with greenery, and offer comfortable living conditions. By extensively exploring subjects such as the advantages, challenges, and even the possibility of misleading environmental claims, this study advocates for a well-informed approach. It emphasizes the importance of making deliberate choices and upholding unwavering commitment to genuine sustainability when incorporating green elements into tall buildings.

Overall, the paper meets its stated four objectives as follows:

1.  Introduction and Awareness: The paper raises awareness about this innovative approach to urban development by offering an overall introduction to greenery-covered tall buildings. Integrating greenery into tall buildings presents a promising solution to enhance the urban environment while addressing pressing environmental concerns. This design direction could be the future of dense areas with limited "horizontal" land.

2.  Innovative Design Concepts: Mapping novel projects has provided valuable insights into innovative ideas and design concepts. Distinguishing the novel projects from "afterthought" greenery additions would allow identifying successful implementations and showcasing their potential as sustainable and transformative architectural solutions. In these projects, integrating greeneries into structures is not a cosmetic treatment to enhance the appearance of the building. It is integral to the design process. The visual expression in this model stems from genuine "green" design objectives to improve environmental and human health. These projects differ from those that sprinkle plants and trees on buildings to make them look cool!

By examining and classifying projects based on their design approaches toward incorporating greenery, the paper showcased the potential of greenery-covered tall buildings as transformative architectural solutions. The novel projects, which purposefully integrated greenery from the outset of their design, demonstrated a commitment to sustainability and a genuine vision for creating green urban spaces.

3.  Social, Environmental, and Economic Benefits: The paper highlighted the multiple advantages of this architectural typology. The examined projects have been shown to positively impact well-being by providing urban residents access to green spaces promoting physical and mental health. Additionally, greenery-covered tall buildings contribute to urban sustainability by improving air quality, reducing the urban heat island effect, and supporting biodiversity. Access to green spaces within tall buildings

allows city dwellers to reconnect with nature, relieving the stresses of urban life. Studies have shown that exposure to green environments can enhance mental health, reduce stress levels, and promote relaxation and cognitive restoration. The presence of greenery within tall buildings creates unique urban sanctuaries, fostering a sense of serenity and tranquility amid the bustling cityscape.

4. Challenges and Solutions: Lastly, exploring the challenges of integrating trees and plants into tall buildings has underscored the importance of careful planning and design and collaboration among architects, engineers, and developers. Construction costs, maintenance considerations, and adherence to building and fire codes are crucial factors that require thorough evaluation to ensure greenery-covered tall buildings' long-term success and sustainability. "Vertical planting" is far more expensive than "horizontal planting". Integrating plants in towers requires complex engineering solutions to support the plants' weight and movement and incorporate irrigation systems. Plant maintenance involves regular pruning, fertilizing, pest control, and replacement. Integrating greeneries and associated irrigation systems in tall buildings requires unique expertise and technical knowledge. Repairing and upgrading these systems and maintaining plants demand operational and maintenance costs. Water costs could also be considerable. We also need to reduce the carbon footprint in constructing these buildings.

Collectively, the paper contributes valuable knowledge to the architectural discourse on greenery-covered tall buildings. The research gives architects, developers, scholars, and the public a deeper understanding of this emerging building typology and its potential to transform urban landscapes into more sustainable, green, and livable spaces. Through discussions of opportunities, challenges, and potential greenwashing, the study encourages stakeholders to approach the implementation of greenery-covered tall buildings with informed decision making and a commitment to genuine sustainability. Integrating nature into urban architecture offers a promising path toward more resilient, vibrant, and environmentally conscious future cities.

## 7. Future Research

High-rise landscaping is still a relatively new concept in the design and development industries. While this paper has provided valuable insights into greenery-covered tall buildings and their potential as an innovative architectural typology, several avenues for future research could further enrich the understanding and development of this concept. The following areas are suggested for exploration:

1. Economic Analysis: A limited amount of financial research and analysis are available on this topic. The return on investment (ROI) for greenery-covered tall buildings is an important aspect that requires more in-depth analysis. Conducting longitudinal studies on existing greenery-covered tall buildings can provide valuable data on these projects' long-term performance and sustainability. Assessing the ecological impact, energy efficiency, and occupant satisfaction over extended periods will help validate the benefits of this building typology.

2. Biophilic Design and Human–Environment Interaction: Investigating the psychological and physiological effects of greenery-covered towers on occupants will contribute to our understanding of biophilic design principles and the impact of nature on human well-being. Future research could delve into the cognitive benefits, stress reduction, and productivity improvements associated with proximity to green spaces within tall buildings.

3. Design Optimization and Adaptation: Exploring design strategies that optimize the integration of greenery into tall buildings can enhance the effectiveness and practicality of such projects. Research into innovative materials, irrigation systems, and plant selection may lead to more sustainable and resilient greenery-covered towers.

4. Comparative Studies: Comparative analyses between greenery-covered towers and conventional tall buildings can help to better understand this architectural typology's

relative advantages and challenges. Evaluating performance metrics, construction costs, and maintenance requirements will help stakeholders make informed decisions regarding greenery integration in future urban projects.

By delving deeper into the abovementioned areas, researchers can further advance our understanding of this innovative building typology and its potential to create greener, more resilient, and socially inclusive urban environments. Continued research and collaboration among architects, developers, and scholars will drive the evolution of greenery-covered towers as a transformative approach to sustainable urban development.

**Funding:** This research received no external funding.

**Data Availability Statement:** Not applicable.

**Acknowledgments:** The author thanks the *Buildings* Journal for its support and the reviewers for valuable feedback.

**Conflicts of Interest:** The authors declare no conflict of interest.

## Appendix A

### Tall buildings

There is no commonly acknowledged definition of the term "tall building". Governments throughout the world define "tall buildings" differently. For instance, German regulations define a "tall building" as a structure higher than 22 m (72 feet) with space for permanent human habitation. City officials derived this restriction from the length of ladders used by firefighters. Leicester City Council in the United Kingdom defines a tall building as any structure over 20 m/66 ft in height. The city of Cork defines towering buildings in Ireland as having ten stories or more [110]. Other cities define them as buildings of any height significantly taller than most of the surrounding area or a building that would significantly alter the city's skyline.

The perception of what constitutes a tall building can vary based on local building regulations, architectural norms, and the urban context of a specific city or region. In some cities or regions, a 10-story building might indeed be considered tall, especially if it significantly rises above the surrounding structures and is distinct in height and appearance. On the other hand, in urban areas with numerous high-rise buildings and skyscrapers, a 10-story building might be considered more of a mid-rise structure.

Integrating greenery vertically in buildings can be more challenging than doing so horizontally, as vertical greenery requires careful consideration of structural and engineering aspects and irrigation and maintenance requirements. Given the challenges of integrated greeneries in higher altitudes, a threshold of 10 stories could be reasonable. Yet, this is a subjective viewpoint, and the classification of a building as "tall" can vary based on different criteria and contexts.

While the number 10 may be considered arbitrary in classifying buildings as tall or high-rise, it serves a valuable purpose in research as it provides a baseline to map buildings of a unique typology, like greenery-covered or vegetated buildings. In research, establishing a baseline is crucial for creating a common framework and reference point for comparison. By using a specific number, such as 10 stories, as the threshold for tall buildings, researchers can differentiate and categorize buildings based on their height relative to this baseline. This helps organize and analyze data, identify trends, and understand the characteristics of buildings within a specific typology.

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
