# Peer review of "Greenery-Covered Tall Buildings: A Review"

_buildings, doi:10.3390/buildings13092362_

Round 1
Reviewer 1 Report
lines 20-28: the incipit is quit generalist.
lines 29-36: the link between Covid-19 and biofilic design approach is not enough supported by scientific evidence. The same is for the statement: "The pandemic of COVID-19 has underlined the need for green and healthy spaces through biophilic design that connects people with Nature [11]".
lines 50-60: the source [23] is not a scientific source, the data can not be quoted in this way.
lines 61-69: same as above.
lines 70-78: this is an announcement, not appropriate.
lines 80-87: data and scientific references would be needed.
lines 111-112: there are several reliable studies that provide analysis and models related to the energy performence of green roofs and also green façades, detecting both performance and inefficiences. The state of the art is evolving, it would be important to quote scientific sources.
The "Method" is not clear. And there is not a replicability setting of it.
All case studies are mentioned superficially: the technical licterature about mentioned solutions exists.
lines 301-317: the benefits of vegetation are largerly investigated by scientific licterature. But the benefits of building element integrated by vegetation is something different: as example, "Preventing soil erosion and water runoff by stabilizing the ground and retaining moisture" (line 305) is not a benefit of a green roof or a green façade, so it is out of contest.
lines 336-343: the Bosco Verticale has 3 different binds for trees. The proper source of information is missed.
lines 333-334; 343-343: this sentence is an obviousness. The issue of weight of vegetative systems is an important topic, but needs to be better framed.
lines 349-364: balconies in tall buildings are analized by relevant studies, not quoted.
lines 373: thise sentence needs a source. This type of calculation exists and are published.
The state of the art regarding Maintenance and Irrigation (Sub-sections of Discussion Section) is quit evolved and also published (as first by international standards and guidelines). The sources are missed.
In general the article is not focused.
The selection of the case study does not follow clear reasons.
There is not an omogeneous analysis/comparison of the case studies.
The choosen aspects of the discussion are not investigated, just declared but in a very general way. As consequence the conclusions are simplified.
The reliable sources are missed (standards, paper, research report, scientific high level studies).
Reviewer 2 Report
Does the introduction provide sufficient background and include all relevant references?
The introduction is well-structured and presents the justification for the proposed objective. The references mentioned are relevant and recent.
Are all the cited references relevant to the research?
Yes, however, the author must verify the correspondence between the citations of authors throughout the text and the listed references.
Is the research design appropriate?
This paper examines “the innovative design approaches of greenery-covered towers, drawing on examples from different countries and contexts”, from the technical, environmental, social and economic point of view. It addresses the Biophilic design. "The Biophilic design is an approach that integrates natural elements and processes into the built environment, enhancing the well-being and resilience of the inhabitants”. Specific objectives are:
“• Offer an overall introduction to the greenery-covered towers.
• Map out major projects that integrate greenery into tall buildings.
• Highlight and examine the “innovative” greenery architectural elements and systems.
• Discuss the opportunities, challenges, and greenwashing of this architectural model.”
This is a very interesting topic from the point of view of sustainable development, especially in the case of densely populated cities with few areas of vegetation. It presents three case studies located in different countries and contexts. The research design is appropriate.
Are the methods adequately described?
Yes, the methods are adequately described. However, the author could have explained their approach to their search for relevant articles more explicitly. For example, could have explained which databases and strings they used.
Are the results clearly presented?
As a result, it presents the details of the three selected case studies and conducts a discussion addressing the advantages and environmental and health benefits of implementing Biophilic design, making a counterpoint with the disadvantages and care in integrating vegetation in tall buildings. In this context, discussions are focused on 4 aspects: Maintenance, Watering, Plants’ Health and Experiencing Nature.
Are the conclusions supported by the results?
Yes, the conclusions are supported by the results they obtained and the discussions are relevant. The paper discusses, in a generalized but interesting way, the advantages, disadvantages, challenges and precautions when carrying out a feasibility study of high-rise buildings, with the insertion of vegetation in their architectural and structural design. Therefore, it is an embryonic paper that will raise further discussions and studies, with an emphasis on the main topics of discussion listed in Chapter 5.
Reviewer 3 Report
Through a detailed analysis of examples of green high-rise architecture, this report demonstrates the benefits of greening at the urban scale, including in the context of climate change and reducing energy consumption.
The discussions in the study provide the reader with interesting insights that can be considered as guidelines for future designs. Some drawbacks are also reported.
The study has a linear and homogeneous structure. The case studies are analysed in detail and the results are discussed in detail.
The conclusions are well described.
I would recommend enriching the text with more illustrations of the case studies.
In the methodology section, the analysis of the scientific literature should be clarified (did it only serve to extrapolate the examples?) as the discussions are only based on the analysis of the case studies.
It would be desirable to link paragraph 2 to the first to avoid repetition and create a more linear discourse.
Reviewer 4 Report
Please see the comments in the attachment.

Round 2
Reviewer 1 Report
lines 20-28: the incipit is quit generalist.
Indeed, as the first paragraph of the Introduction, following the Abstract, it is acceptable to have introductory, general information, paving the way to more specific information.
Anyway, I have revised the first page. Please see lines 26-77.
Ok
lines 29-36: the link between Covid-19 and biofilic design approach is not enough supported by scientific evidence. The same is for the statement: "The pandemic of COVID-19 has underlined the need for green and healthy spaces through biophilic design that connects people with Nature [11]".
The reviewer misread these lines. They are not meant to establish a scientific correlation between Covid-19 and biophilic design. Instead, it is meant to show how Covid-19 has reinforced our desire to connect with nature.
Anyway, I have revised the first page. Please see lines 26-77.
I have also added a sub-heading for biophilic design, lines 79-102.
Ok
lines 50-60: the source [23] is not a scientific source, the data can not be quoted in this way.
Disagree. Reference 23 points to the official website of Stefano Boeri Achitetti, who is the responsible architect for the discussed project (Liuzhou Forest City). There is no better reference than the one provided. There is no can provide information about this project than his office, the designers. All other references are secondary ones.
In the link of the reference [23]: https://www.stefanoboeriarchitetti.net/en/project/dubai-vertical-forest/
I can not find the data written in the lines 50-60 ver.1; lines 143-149 ver.2:
According to Boeri’s website, the project claims to absorb almost 10,000 tons of CO2 and 57 tons of microparticles annually while concurrently creating approximately 900 tons of oxygen, assisting in the fight against severe problems associated with air pollution. It promises to absorb nearly 10,000 tons of CO2 and 57 tons of microparticles yearly, simultaneously producing about 900 tons of oxygen, thereby combating severe air pollution problems [23].
Anyway, I wonder what are the sources of Stefano Boeri website. What are the scientific sources used by Stefano Boeri?
It is plausible that the architect (Stefano Boeri) is interested in divulging positive data regarding the presumed performance/benefits of his projects. If there are no scientific studies to support the quoted data, this citation is not acceptable. I appreciate that the author wrote "According to Boeri’s website […]” but I believe it would be correct to quote scientific study, from scientific journals, regarding maybe single trees performance (CO2 absorption, oxygen production, etc.), or specific species, characterized by defined characteristics and dimensions…
lines 61-69: same as above.
Same as above. These lines discuss a project (Dubai Vertical Tower) by Architect Stefano, and his source is the most accurate source of information.
lines 70-78: this is an announcement, not appropriate.
Disagree. This is more than an announcement. It is an actual project taking place in the real world. The Council on Tall Buildings and Urban Habitat has verified this project.
Anyway, the written source is not Ctbuh. The Skyscrapercenter (from Ctbuh) source is: https://www.skyscrapercenter.com/building/sth-bnk-by-beulah-tower-1/33881
lines 80-87: data and scientific references would be needed.
I have placed references, lines 173.
Ok
lines 111-112: there are several reliable studies that provide analysis and models related to the energy performence of green roofs and also green façades, detecting both performance and inefficiences. The state of the art is evolving, it would be important to quote scientific sources.
This article engages with qualitative discourse discussing design ideas and it is not engaged into quantitative analysis. Please see first paragraphs in the Methods Section, lines 243-265
It is the author's choice to mention the performance of the greenery technologies (which I share).
Lines 243-260 ver.2: this part is not part of the Methods section and it is also vague, with not distinctions about the different performance and benefits related to the green envelope (Energy performance, benefit of plants, etc.). Anyway this part of the Methods section is related to performance and not the investigation of the paper.
The "Method" is not clear. And there is not a replicability setting of it.
I have revised the Methods Section; please see lines 242-445.
ok
All case studies are mentioned superficially: the technical licterature about mentioned solutions exists.
lines 301-317: the benefits of vegetation are largerly investigated by scientific licterature. But the benefits of building element integrated by vegetation is something different: as example, "Preventing soil erosion and water runoff by stabilizing the ground and retaining moisture" (line 305) is not a benefit of a green roof or a green façade, so it is out of contest.
I have revised the Discussion Section to be more comprehensive balancing the benefits and challenges of this building typology. Please see lines 702-1242. I have also added four case studies.
I agree about line 305, which I removed.
ok
lines 336-343: the Bosco Verticale has 3 different binds for trees. The proper source of information is missed.
No. I am citing ARUP, the engineering company in charge – it is the direct, most accurate, and reliable source.
See Ctbuh research book, free download: https://store.ctbuh.org/research-reports/49-vertical-greenery-2015.html
lines 333-334; 343-343: this sentence is an obviousness. The issue of weight of vegetative systems is an important topic, but needs to be better framed.
Yes, these are known facts but they need to be mentioned as a reminder to support the presented argument.
lines 349-364: balconies in tall buildings are analized by relevant studies, not quoted.
I have one reference and added two references.
lines 373: thise sentence needs a source. This type of calculation exists and are published.
The source is in place.
The state of the art regarding Maintenance and Irrigation (Sub-sections of Discussion Section) is quit evolved and also published (as first by international standards and guidelines). The sources are missed.
In general the article is not focused.
Disagree. The sources are adequate.
The selection of the case study does not follow clear reasons.
I have revised the Methods Section; please see lines 242-445.
ok
There is not an omogeneous analysis/comparison of the case studies.
This is out of the scope of the study, please see the study’s objectives. Future Studies Section has been added suggesting doing a comparative analysis.
ok
The choosen aspects of the discussion are not investigated, just declared but in a very general way. As consequence the conclusions are simplified.
Other reviewers praised the Discussion Section: Reviewer 3: “The case studies are analysed in detail and the results are discussed in detail.” Reviewer 4 explains: “The topic of the paper is up-to-date. It fits in with the concept of green architecture in line with sustainable development. The paper includes many interesting insights, especially in the "Discussion" section.”
Anyway, I have expanded the Discussion Section. Please see lines 702-1242.
I don’t agree sorry. Anyway the paper has improved.
The reliable sources are missed (standards, paper, research report, scientific high level studies).
Untrue. The paper included 88 references, covering the most essential sources on the topic.
The revised copy added more references totaling 110.
5.1.8 According to the different vertical green systems, several green façades “abuse” of water. In any case a green façade is able to intercept rain water, so green façades do not operate rain water management, differently from the green roofs.
The Discussion section - in particular 5.1 Benefits and subsections, but in part also 5.2 Challenging and subsections - is not linked to the scope of the paper Greenery-Covered Tall Buildings: Examining Innovative Design Approaches:
5.1 try to summarize again performance and benefits;
5.2 looks like more in line with the spirit of the paper… but just some sub-sections are focused.
Author Response
Please see the attached response.

Reviewer 4 Report
The paper has been revised in a satisfactory manner. It can be published in the present form as a review article.
Author Response
Thanks very much for the positive response:
"The paper has been revised in a satisfactory manner. It can be published in the present form as a review article."
Round 3
Reviewer 1 Report
The paper has improved.